

# Leipzig Ice Nucleation chamber Comparison (LINC): Inter-comparison of four online ice nucleation counters

Monika Burkert-Kohn[1], Heike Wex[2], André Welti[2], Susan Hartmann[2], Sarah Grawe[2], Lisa Hellner[2], Paul Herenz[2], James D. Atkinson[1], Frank Stratmann[2], and Zamin A. Kanji[1]

[1]Institute for Atmospheric and Climate Science, ETH Zurich, Universitaetstrasse 16, 8092 Zurich, Switzerland
[2]Leibniz-Institute for Tropospheric Research (TROPOS), Permoserstrasse 15, 04318 Leipzig, Germany

*Correspondence to:* Monika Burkert-Kohn (Monika.Burkert@env.ethz.ch) and Zamin A. Kanji (Zamin.Kanji@env.ethz.ch)

**Abstract.** Ice crystal formation in atmospheric clouds has a strong effect on precipitation, cloud lifetime, cloud radiative properties and thus the global energy budget. Primary ice formation above 235 K is initiated by nucleation on seed aerosol particles called ice nucleating particles (INPs). Instruments that measure the ice nucleating potential of aerosol particles in the atmosphere need to be able to accurately quantify ambient INP concentrations. In the last decade several instruments have been
developed to investigate the ice nucleating properties of aerosol particles and to measure ambient INP concentrations. Therefore, there is a need for inter-comparisons to ensure instrument differences are not interpreted as scientific findings.

In this study, we inter-compare the results from parallel measurements using four online ice nucleation chambers. Seven different aerosol types are tested including untreated and acid treated mineral dust (microcline - a K-feldspar - and kaolinite), as well as birch pollen washing waters. Experiments exploring heterogeneous ice nucleation above and below water saturation are
performed to cover the whole range of thermodynamic conditions that can be investigated with the inter-compared chambers. The Leipzig Aerosol Cloud Interaction Simulator (LACIS) and the Portable Immersion Mode Cooling chAmber coupled to the Portable Ice Nucleation Chamber (PIMCA-PINC) performed measurements in the immersion freezing mode. Additionally two continuous flow diffusion chambers (CFDCs) PINC and the Spectrometer for Ice Nuclei (SPIN) are used to perform measurements below and just above water saturation nominally presenting deposition nucleation and condensation freezing.

The results of LACIS and PIMCA-PINC agree well over the whole range of measured frozen fractions ($FF$s) and temperature. In general PINC and SPIN compare well and the observed differences are explained by ice crystal growth and different residence times in the chamber. To study the mechanisms responsible for ice nucleation in the four instruments, $FF$ (from LACIS and PIMCA-PINC) and activated fraction, $AF$ (from PINC and SPIN) are compared. Measured $FF$s are up to a factor of three higher than $AF$s, but not consistent for all aerosol types and temperatures investigated. It showed that measurements
from CFDCs cannot be assumed to produce the same results as those instruments exclusively measuring immersion freezing. Instead the need to apply a scaling factor to CFDCs operating above water saturation has to be considered to allow comparison with immersion freezing devices. Our results provide further awareness on factors such as the importance of dispersion methods and the quality of particle size-selection for inter-comparing online INP counters.



# 1 Introduction

Ice crystal formation in the atmosphere changes cloud physical and optical properties, thus influencing the lifetime of clouds and is important for precipitation formation (Lohmann and Feichter, 2005). Ice nucleation mechanisms and the properties of aerosol particles acting as so-called INPs, which are seed particles necessary for ice nucleation to occur on, are not sufficiently

understood and demand further investigation to accurately parameterize atmospheric ice formation in models for weather and climate. Laboratory measurements on well-characterized aerosol particles and ambient observations improve our understanding of atmospheric ice nucleation processes and INP abundance in the atmosphere. This helps to quantify the role of different INPs on cloud formation and the conditions commonly found in the atmosphere.

Ice nucleation can occur via different mechanisms, either homogeneously at temperatures ($T$) lower than $235\,\mathrm{K}$ (e.g., Prup-

pacher and Klett, 1997), or heterogeneously − catalyzed by an INP which provides a surface for ice to nucleate on at $T > 235\,\mathrm{K}$. For heterogeneous ice nucleation, several pathways are distinguished: Deposition nucleation, in which water vapor directly deposits on an INP to form ice in water subsaturated conditions; contact freezing where ice formation is due to a supercooled cloud droplet colliding with an INP; condensation freezing in which water vapor directly deposits on an INP to form water and/or ice in water supersaturated conditions with the existence of liquid water expected but not explicitly ob-

served; and immersion freezing, where an INP is immersed in a droplet and has attained sufficient supercooling to freeze (e.g., Vali, 1985). While the above given descriptions are followed in this paper, it is discussed that there may not be a difference between condensation and immersion freezing on a process level, when possible freezing point depressions are accounted for (Wex et al., 2014; Vali et al., 2015). Further, Marcolli (2014) suggested that deposition nucleation might in fact be immersion freezing of water trapped in pores and cavities at water subsaturated conditions. Which ice nucleation pathways exist and under

which conditions they are relevant in the atmosphere is not fully understood. However, there is no reason to believe that in the atmosphere, and particularly in mixed-phase clouds, a difference in the freezing mechanism might be relevant, as ice formation in these clouds generally proceeds via the liquid phase.

Instruments developed to explore ice nucleation for different formation pathways and to measure the concentration of atmospheric INPs fall into two broad categories: Offline measurements of aerosol collected on filters (e.g., Bigg, 1967; Klein et al.,

2010; Conen et al., 2012) or in suspensions (e.g., Hader et al., 2014), which operate on hour to day timescales, and online measurements which are capable of real-time detection of INP concentration with a higher temporal resolution from seconds to minutes. The portable online instruments report INP concentrations for both ground-based (e.g., DeMott et al., 2010; Chou et al., 2011; Garcia et al., 2012; Tobo et al., 2013) and airborne measurements (Rogers et al., 2001a; DeMott et al., 2003a, b, 2010).

Inter-comparing instruments in the laboratory under controlled conditions is necessary to characterize their performance for field studies and to compare quantitative reproducibility. Some studies have already investigated the comparability of a number of online and offline instruments on selected aerosol types in a laboratory setting (DeMott et al., 2011; Wex et al., 2014, 2015; Hiranuma et al., 2015). Ice nucleation measurements have been conducted with the Leipzig Aerosol Cloud Interaction Simulator (LACIS, Hartmann et al., 2011, immersion mode) in parallel to the Colorado State University-Continuous Flow



Diffusion Chamber (CSU-CFDC, Rogers et al., 2001b, condensation mode) on size-selected kaolinite particles including samples coated with soluble material (Wex et al., 2014). Lower $FF$s were measured in LACIS compared to the CSU-CFDC, and good agreement was found when particle residence times in the respective instruments were accounted for. Wex et al. (2015) reported measurements on size-selected Snomax® particles with seven instruments including LACIS and the Portable Ice Nu-

cleation Chamber (PINC, Chou et al., 2011). Wex et al. (2015) examined droplets formed on single particles and droplets taken from suspensions containing Snomax®. The varying mass of Snomax® per droplet or per particle that was examined by the different instruments was accounted for, and the results agreed within a factor of three below 263 K for all instruments. The PINC measurements at water supersaturated conditions showed an activation onset temperature (ice fraction larger $\sim 10^{-3}$) of 2 K lower than LACIS i.e. less ice activity was observed in PINC than for immersion freezing with LACIS. What factors

cause this deviation is not yet explored. DeMott et al. (2015) presented a comparison of a CFDC to the Aerosol Interaction and Dynamics in the Atmosphere (AIDA) cloud chamber and found agreement only when an empirically determined factor of three was applied to the CFDC data for their measurements of mineral dust at a relative humidity with respect to water ($RH_{\rm w}$) of 105 %. A previous study on immersion freezing (Hartmann et al., 2016) observed differences in the ice nucleation active site density ($n_{\rm s}$) between LACIS and the Immersion Mode Cooling chAmber coupled to the Zurich Ice Nucleation

Chamber (IMCA-ZINC, Lüönd et al., 2010) when the same particle type and size was tested at different times/locations. An offset of about one order of magnitude in $n_{\rm s}$ or a temperature shift of $5-6$ K for kaolinite particles was found. The fraction of multiple-charged particles as a reason for the deviation was discussed in Hartmann et al. (2016). For the study of Hiranuma et al. (2015) an identical illite NX sample was used for their comprehensive inter-comparison of 17 ice nucleation instruments which showed an even larger deviation of 8 K or three orders of magnitude in $n_{\rm s}$ between the different instruments. So far it

has not been possible to narrow down whether these discrepancies are inherent to the instruments used or other factors such as the particle generation techniques and size-selection are the cause, because a number of instruments in the Hiranuma et al. (2015) study were not operated in parallel. The discrepancies found in the Fourth International Ice Nucleation workshop with a selection of different instruments and aerosol types (e.g., DeMott et al., 2011; Kanji et al., 2011) emphasized the importance of parallel measurements for a direct comparison of ice nucleation instrumentation. Parallel measurements of specific particle

sizes can be used to identify any discrepancies that arise from the ice nucleation methods itself while excluding factors such as differences in aerosol sample, particle generation method or particle size.

We present a comparison of four online ice nucleation instruments performed during the Leipzig Ice Nucleation chamber Comparison (LINC) in September 2015, which was hosted by TROPOS, Leipzig. Seven different types of size-segregated aerosol particles were tested for their immersion freezing potential, four were additionally tested for condensation freezing and

deposition nucleation. The samples were two mineral dusts (microcline and kaolinite), nitric or sulfuric acid treated microcline particles and birch pollen washing water of samples from the Czech Republic and Sweden. During long-range transport of aerosol particles in the atmosphere, coating with e.g. sulfuric acid can cause a temporary or permanent change in the physicochemical properties of the particles and can decrease their ice nucleation activity. Acid treatment of microcline particles was chosen in the present study to investigate a permanent change after treatment and removal of the acid coating. The selec-

tion of aerosol types known to be ice active at different temperatures allows for comparison over the full range of detectable





frozen/activated fractions possible with the instruments.

A simultaneous comparison of LACIS and PIMCA-PINC (Portable Immersion Mode Cooling chAmber coupled to PINC) as well as the direct comparison between PINC and SPIN (the Spectrometer for Ice Nuclei) on size-selected aerosol particles in water sub- and supersaturated conditions is presented for the first time allowing to investigate instrument specific differences.

Furthermore, observations with the four instruments are used for an explicit comparison of immersion freezing of droplets containing a single aerosol particle (LACIS and PIMCA-PINC) to experiments using dry particles above water saturation, where it is not possible to distinguish between immersion and condensation freezing (SPIN and PINC).

## 2 Materials and methods

### 2.1 Aerosol samples and treatment

The kaolinite sample used in this study is a commercially available product from Fluka (same as Sigma-Aldrich). The microcline sample is a K-feldspar from Minas Gerais in Brazil consisting of $76\%$ microcline (K-feldspar) and $24\%$ albite (Na-feldspar) (Augustin-Bauditz et al., 2014). It was provided by the Technical University Darmstadt within the framework of the Ice Nucleation research UnIT (INUIT). $2.5\,\mathrm{g}$ of the respective powder material was suspended in $30\,\mathrm{ml}$ of double deionized water (Milli-Q, $18.2\,\mathrm{M\Omega cm}$) for the purpose of wet aerosolization. For the acid treatment, $2.5\,\mathrm{g}$ microcline powder was suspended

in $30\,\mathrm{ml}$ of 1M sulfuric or nitric acid solution for about twelve hours. To remove the acid, the suspension was centrifuged at $17000\,\mathrm{rpm}$ for ten minutes to settle particles. The supernatant was removed from the sample, its pH level determined, and the sample diluted with Milli-Q water. This step was repeated several times until the pH of the supernatant reached the pH of deionized water (pH $\sim 5$). The pollen washing water was made from two birch pollen samples belonging to the species *Betula pendula*. One birch pollen sample originated from the Czech Republic (Pharmallerga®, referred to as birchS) and the

other one from Sweden (AllergonAB®, referred to as birchN). The sample preparation of the pollen washing water followed the procedure described in Pummer et al. (2012). One gram of the pollen was suspended in $20\,\mathrm{ml}$ of Milli-Q water. After one night (about $12\,\mathrm{h}$) in the refrigerator, the pollen grains were removed from the suspension by gravitational filtering (round filter, Schleicher and Schüll Selecta 595, pore size $4-7\,\mathrm{\mu m}$).

### 2.2 Instrumental setup, particle generation and size-selection

A schematic of the instrumental setup is shown in Fig. 1. All samples were aerosolized from suspension using a home-built atomizer (design similar to TSI, Model 3076). Droplets of the suspension were passed through a diffusion dryer creating agglomerates from the residuals of the droplets, which were size-selected in a Differential Mobility Analyser (DMA, type Vienna medium, Knutson and Whitby, 1975). To remove multiple-charged larger particles, a cyclone ($D_{50} = 500\,\mathrm{nm}$) was operated at $4\,\mathrm{l min^{-1}}$ downstream of the DMA. As shown in Fig. 1, downstream of the cyclone an aerosol distributor (mixing

volume) supplied the aerosol ($RH_\mathrm{w}$ below $1\%$) to all instruments including a Condensation Particle Counter (CPC, TSI, Model 3010), a Cloud Condensation Nucleus Counter (CCNC, Droplet Measurement Technologies, Roberts and Nenes, 2005), an



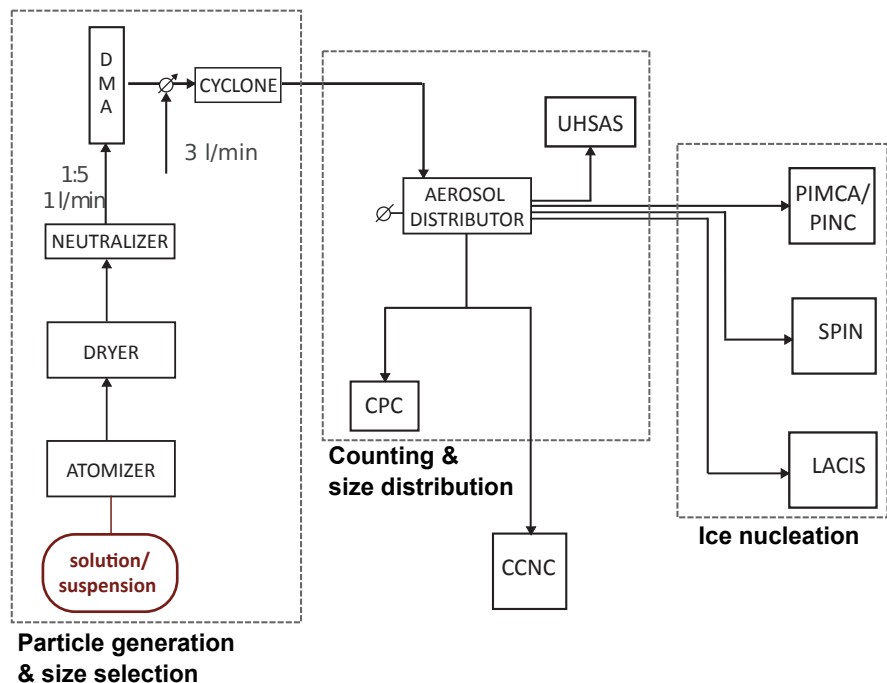

**Figure 1.** Schematic of the inter-comparison setup during LINC. Components include a Differential Mobility Analyzer (DMA), Condensation Particle Counter (CPC), Cloud Condensation Nucleus Counter (CCNC), Ultra High Sensitivity Aerosol Spectrometer (UHSAS), the Leipzig Aerosol Cloud Interaction Simulator (LACIS), the Portable Immersion Mode Cooling chAmber (PIMCA), the Portable Ice Nucleation Chamber (PINC) and the Spectrometer for Ice Nuclei (SPIN).

Ultra High Sensitivity Aerosol Spectrometer (UHSAS, Droplet Measurement Technologies), SPIN, LACIS and either PINC or PIMCA-PINC (see Sect. 2.3 for description of ice nucleation chambers). Individual impactors upstream of the ice counters were not used for any of the experiments to exclude biases from particle losses. UHSAS measurements indicated a substantial reduction in the number of multiple-charged particles by the cyclone but not complete removal. Table 1 summarizes the

5 selected particle sizes and the fraction of multiple-charged particles based on UHSAS measurements during the ice nucleation experiments. It was found that the fraction of multiple-charged particles remained constant in time during each experiment for all particle types investigated. Therefore, the measurements of identical samples at different times have been averaged. CCNC measurements were used to derive particle hygroscopicities. Hygroscopicities of acid treated and untreated particles were similar (with the treated sample having a slightly lower CCN activity) indicating that soluble material added during the acid

10 treatment was completely removed by the applied procedure of repeated rinsing of the sample. Typical particle concentrations during the ice nucleation experiments measured with a CPC in parallel were $240 \pm 70 \, \text{cm}^{-3}$ for the presented measurements and diluted to $25 - 40 \, \text{cm}^{-3}$ for PIMCA-PINC measurements.





**Table 1.** Size and fraction of single- (1) or multiple-charged particles (2, 3 and 4) of the resulting aerosol after size-selection and the cyclone.

| Aerosol type | | Charges | | | |
|---|---|---|---|---|---|
| | | 1 | 2 | 3 | 4 |
| Microcline | Stokes size [nm] | 200 | 324 | 439 | 552 |
| | Fraction | 0.593 | 0.307 | 0.086 | 0.014 |
| Microcline | Stokes size [nm] | 300 | 507 | 706 | |
| | Fraction | 0.815 | 0.182 | 0.003 | |
| Microcline $H_2SO_4$ | Stokes size [nm] | 300 | 507 | 706 | |
| | Fraction | 0.821 | 0.162 | 0.017 | |
| Microcline $HNO_3$ | Stokes size [nm] | 300 | 507 | 706 | |
| | Fraction | 0.867 | 0.131 | 0.002 | |
| Kaolinite | Stokes size [nm] | 500 | 889 | | |
| | Fraction | 0.988 | 0.012 | | |
| BirchS | Stokes size [nm] | 500 | 889 | | |
| | Fraction | 0.935 | 0.065 | | |
| BirchN | Stokes size [nm] | 300 | 507 | 706 | |
| | Fraction | 0.893 | 0.105 | 0.002 | |

## 2.3 Description of ice nucleation chambers

### 2.3.1 PINC

PINC is a portable parallel-plate vertical CFDC with two individually temperature controlled walls. Prior to an experiment, a thin ice-layer is applied to the chamber walls to provide a source of water vapor. A difference in temperature ($\triangle T$) is set

between the walls that generates a parabolic supersaturation profile with a peak saturation close to the center plane. Sample aerosol is introduced with a flow rate of $1\,\mathrm{l\,min^{-1}}$ and layered between two particle-free sheath air flows ($4.5\,\mathrm{l\,min^{-1}}$ on each side) ensuring a narrow, centered sample lamina. After a residence time ($t_{\mathrm{res}}$) of $4-5\,\mathrm{s}$ in the ice nucleation section, the aerosol enters the evaporation section of the chamber where both walls are isothermally set to the warm wall temperature. In the subsaturated environment with respect to liquid water any formed droplets evaporate while ice crystals are maintained at

ice saturated conditions until detection. At the bottom of the chamber, exiting aerosol particles and ice crystals are counted by an optical particle counter (OPC, Lighthouse R5104). Particles larger than a set size threshold are counted as ice crystals. For data in this study a size threshold of $2\,\mu\mathrm{m}$ (diameter) is used. Ice nucleation below water saturation ($RH_{\mathrm{w}} < 100\,\%$) is classified as deposition nucleation and above water saturation ($RH_{\mathrm{w}} \geq 100\,\%$) as condensation freezing. The accuracy of the temperature sensors is $\pm 0.1\,\mathrm{K}$ and the variation of temperature across the sample lamina $\pm 0.4\,\mathrm{K}$. This corresponds to an

uncertainty in $RH_{\mathrm{w}}$ of $\pm 2\,\%$ (Chou et al., 2011). Experiments consist of a scan in $RH$ at a prescribed $T$ and are conducted





from ice saturation to above water saturation up to an $RH$ at which droplets cannot be distinguished from ice crystals based on size (droplet breakthrough). Before and after each scan, background concentrations of ice crystals in the chamber are obtained while sampling filtered air. Background counts are linearly interpolated between two filter periods and subtracted from the sample signal. The activated fraction ($AF$) is calculated as the ratio of ice crystals detected with the OPC to the number of

total aerosol particles measured with the CPC. Uncertainty in $AF$ is $14\%$, resulting from $10\%$ uncertainty in each the OPC and CPC measurements. More details on the design and operation of PINC can be found in Chou et al. (2011).

### 2.3.2   PIMCA-PINC

The PIMCA-PINC setup is the portable version of the laboratory design IMCA-ZINC (Lüönd et al., 2010; Stetzer et al., 2008) allowing for measurements explicitly in the immersion freezing mode. PIMCA is a vertical extension of PINC in which aerosol

particles are activated to cloud droplets at $303\,\mathrm{K}$, prior to supercooling the droplets to the desired ice nucleation temperature. $RH_{\mathrm{w}}$ in PINC is set to water saturation conditions to maintain cloud droplets at a radius of $5 - 7\,\mathrm{\mu m}$. Flow rates are set to $0.6\,\mathrm{l\,min^{-1}}$ sample air with $2.2\,\mathrm{l\,min^{-1}}$ of sheath air on either side of the aerosol lamina. This gives a residence time of $\sim 7\,\mathrm{s}$ at ice nucleation conditions in PINC. Ice crystals and cloud droplets are distinguished via depolarization with the ice optical detector IODE (Nicolet et al., 2010). Unlike the detection system used in the PINC configuration, IODE only observes a small

volume of the sample lamina. The frozen fraction ($FF$) is derived from the ratio of ice crystals to the total particles detected in this subset of the sample. More details on the specifications of the PIMCA-PINC setup can be found in Kohn et al. (2016). In a typical experiment a temperature scan is performed, starting at homogeneous freezing conditions at $T < 233\,\mathrm{K}$. Temperature is then increased until the detected $FF$ is not distinguishable anymore from the experimental background. Each reported data point consists of an average of two to five individual measurements at the same $T$. This adds up to more than 3000 individual

(particle) intensity peaks analyzed per data point shown. Error bars in $FF$ indicate the measurement uncertainty from the classification of ice crystals and cloud droplets and the statistical error using standard error propagation. The temperature uncertainty is $\pm 0.4\,\mathrm{K}$ due to variation across the sample lamina and accuracy in the thermocouples of $\pm 0.1\,\mathrm{K}$.

### 2.3.3   SPIN

The SPIN geometry is equivalent to PINC but with a longer ice nucleation section allowing roughly double the residence

time. It is the first commercially available ice nucleation chamber (Droplet Measurement Technologies, Inc.) and was recently described by Garimella et al. (2016). Particle residence time in the ice nucleation section is $9 - 12\,\mathrm{s}$ depending on the $T$ and $RH$ set points of the experiment. Additionally, experimental temperatures as low as $228\,\mathrm{K}$ can be reached and the temperature and supersaturation conditions in the evaporation section can be controlled independently from the ice nucleation section. Similar to PINC, ice crystals are discriminated from non-activated aerosol particles by a size threshold. For this study a threshold size

of $2.5\,\mathrm{\mu m}$ is used. This threshold size is somewhat larger compared to PINC and chosen to clearly distinguish ice crystals from background counts. For SPIN, the chamber background is determined at the beginning of each $RH$ scan (relative humidity with respect to ice ($RH_{\mathrm{i}}$) below $103\%$) while sampling aerosol resulting in a few aerosol counts in the ice channel. This signal is subtracted from the ice counts detected during the experiment. The $AF$ is obtained in the same way as for PINC. Uncertainty




in $AF$ is $14\%$ due to a $10\%$ uncertainty in both the SPIN OPC and the CPC. Temperature uncertainties give the highest and lowest deviation from the average lamina temperature to the calculated temperature between 15 opposite pairs of temperature measurements along the walls of SPIN. Experimental uncertainties are typically within $\pm 1\,\mathrm{K}$ for temperature and $\pm 5\%$ for supersaturation as reported for homogeneous freezing experiments by Garimella et al. (2016).

### 2.3.4  LACIS

LACIS (Hartmann et al., 2011) is a laminar flow tube where, in contrast to ice coated CFDCs, humidified sheath air is the source of water vapor and the ice covered tube walls are water vapor sinks. LACIS consists of seven one-meter long tube sections with an internal diameter of $15\,\mathrm{mm}$ with each tube section separately temperature controlled by a thermostat. The aerosol surrounded by humidified particle free sheath air enters LACIS in an isokinetic fashion. This leads to the formation of a particle beam with a diameter of roughly $2\,\mathrm{mm}$ at the center of the flow tube. All particles moving along the center-line of the laminar flow tube experience the same humidity and temperature conditions, which depend on the inlet dew point and temperature as well as the wall temperature of the tube sections. Detailed information about the setup can be found in Hartmann et al. (2011). In this study LACIS was operated in the immersion mode. Aerosol particles are activated to droplets which subsequently may freeze upon further cooling while passing the tube. At the LACIS outlet the ratio of frozen droplets to the total droplet number is determined after a residence time of $1.6\,\mathrm{s}$ at the coldest adjusted temperature. Experimental $FF$ is derived from measurements with the Thermo-stabilized Optical Particle Spectrometer for the detection of ice (TOPS-Ice, Clauss et al., 2013) installed underneath LACIS evaluating a change in polarization to distinguish between frozen and unfrozen droplets. For each data point, typically 2000 droplets are examined. The measurement uncertainty in $FF$ is based on counting statistics of TOPS-Ice and the Poisson error is given as twice the standard deviation. The temperature error is $\pm 0.3\,\mathrm{K}$.

## 3  Results and discussion

### 3.1  Results of immersion freezing measurements

A total of seven aerosol samples were investigated for immersion freezing during LINC. Figure 2A summarizes size-segregated measurements conducted with PIMCA-PINC and LACIS for mineral dusts (top) and pollen washing waters (bottom). For comparison fit lines to LACIS data from literature derived with the Soccer Ball Model (SBM) are shown in Fig. 2B/C, which assume external mixtures for particles of different sizes (i.e. being differently charged) as determined for this study (Table 1) and account for the instrumental residence times in PIMCA-PINC ($t_{\mathrm{res}} \sim 7\,\mathrm{s}$) and LACIS ($t_{\mathrm{res}} \sim 1.6\,\mathrm{s}$). Parameters used for the SBM calculations are taken from Niedermeier et al. (2015), Augustin et al. (2013), Augustin-Bauditz et al. (2016) and Hartmann et al. (2016) (see Appendix A and Table A1 for more details).

For 200 and $300\,\mathrm{nm}$ microcline particles (Fig. 2, teal and orange) a good agreement between the two instruments is observed. The temperature at which half the cloud droplets freeze ($T_{50}$) is observed to be $244 - 244.5\,\mathrm{K}$ for PIMCA-PINC and $244 - 245.5\,\mathrm{K}$ for LACIS. The increase in $FF$ with decreasing $T$ is well reproduced by the model calculations for PIMCA-





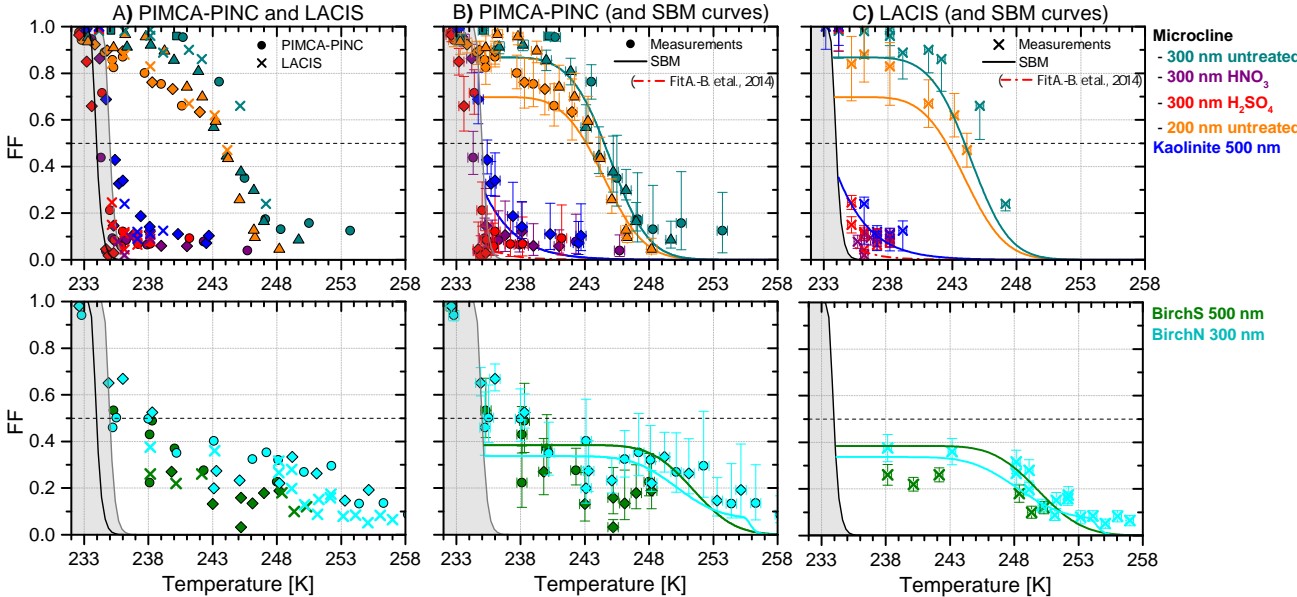

**Figure 2.** Summary of immersion mode experiments. A) Results are shown for untreated and either nitric or sulfuric acid treated mineral dusts (upper panel) and for birch pollen washing waters of two sources (birchS and birchN, lower panel) for PIMCA-PINC and LACIS. The grey and black curves represent homogeneous freezing for PIMCA-PINC (Kohn et al., 2016) and LACIS respectively. $T_{50}$ where half of the droplets have frozen, is indicated by the horizontal black dashed line; different filled symbols correspond to independent PIMCA-PINC experiments. Columns B and C: Curves (except red) show results of the SBM using fit parameters from literature (see text for details) for comparison to data obtained with PIMCA-PINC (B) and LACIS (C) in this study. The red curve is taken from Augustin-Bauditz et al. (2014) representing a fit line to measurements of acid coated mineral dust. Error bars show the uncertainty due to the ice identification technique and the uncertainty in $T$ in each instrument.

PINC (Fig. 2B), but for $T < 240\,\text{K}$ the model results underestimate the measurements, which do not show a pronounced plateau i.e., $FF$ being constant with decreasing $T$, as modeled by the SBM. A plateau of this kind was also not observed for experiments on other untreated mineral dusts in previous studies using similar instrumentation (e.g., Lüönd et al., 2010; Welti et al., 2012; Kohn et al., 2016). Niedermeier et al. (2015) explain the appearance of a plateau with the assumption that not all

5  particles immersed in a droplet feature an ice nucleating site. As the ice activity of particles scales with the surface area and hence with particle size, the height of this plateau scales with particle size, too, and vanishes for sufficiently large particles. For LACIS the SBM curves underpredict the $FF$ and predict $T_{50}$ of $1-2\,\text{K}$ lower than the measurement (Fig. 2C). However, the majority of the results are within measurement uncertainty and a plateau is more apparent in the LACIS data. It is currently unclear why there is a plateau in the LACIS data, but not in the PIMCA-PINC results. Also it should be mentioned that the

10 used SBM parameterization (Niedermeier et al., 2015) is obtained for dry dispersed particles, while particles examined in this study were dispersed from suspensions. A lowering of the ice activity of feldspars when it was suspended and kept in water for





some months prior to ice nucleation measurements is possible. Such effect has been observed in previous immersion freezing studies (Peckhaus et al., 2016; Harrison et al., 2016) and a decrease in $T_{50}$ of $2\,\mathrm{K}$ for the microcline sample used in our study was found by Peckhaus et al. (2016). This could hint towards the lower ice activity in the current study to be an effect of the dispersion method. However, it should be added that the suspensions used in this study were at maximum two weeks old at time

of measurement. Here, both the measured and modeled data are those for the aerosol including multiple-charged, i.e. larger, particles, which allows direct comparison between our measurements and the SBM model. For completeness, the corrected frozen fractions accounting for multiple-charged particles ($FF_{\mathrm{corr}}$) are given in Appendix B and it can be seen that due to the comparably low fractions of multiple-charged particles, differences between uncorrected and corrected values are not large. However, whenever comparing $FF$ to literature data, a possible effect of multiple charges has to be kept in mind.

When microcline samples were treated with either sulfuric or nitric acid (Fig. 2, purple and red) the resulting $FF$ is significantly lower and heterogeneous freezing is not quantifiable with PIMCA-PINC due to measurement uncertainties for $T > 235\,\mathrm{K}$. In a previous study Augustin-Bauditz et al. (2014) found a significant decrease in the ice nucleation ability of the same particle type in the immersion mode when coated with sulfuric acid, but without removing the acid prior to the ice nucleation experiment. A fit curve to their data of sulfuric acid coated mineral dusts is presented in Fig. 2B/C as a red curve. A similar processing

("weathering") of feldspars with acids has been indicated in some previous studies to form clay minerals (e.g., Zhu et al., 2006). It is noteworthy that acid treated microcline has a slightly lower freezing curve than kaolinite (Fig. 2, blue). The kaolinite (Fluka) sample in the current study contains of about $5\,\%$ potassium feldspar (Atkinson et al., 2013), which could explain the higher $FF$ compared to the acid treated microcline. This study shows that there is a persistent reduction in the ice activity even after removing acid residuals from the microcline surface and it implies altering of the microcline surface properties responsi-

ble for its ice nucleation ability. Formation of a clay mineral shell covering the microcline surface could result in a similar ice nucleation ability of kaolinite and acid treated microcline. Homogeneous freezing is observed at lower temperatures for acid treated microcline samples compared to homogeneous freezing experiments by Kohn et al. (2016, grey area in Fig. 2A/B). A reduction in the hygroscopicity of the aerosol particles due to the acid treatment and washing (see Sec. 2.2) could have led to a delayed droplet activation in PIMCA leading to smaller droplets causing lower $T$ of homogeneous freezing.

Kaolinite particles of $500\,\mathrm{nm}$ (untreated) were less ice active than untreated microcline particles but similar in activity to the acid treated particles. Heterogeneous freezing between $235 - 243\,\mathrm{K}$ with PIMCA-PINC and $236 - 239\,\mathrm{K}$ with LACIS is observed, but a $T_{50}$ was only reached at homogeneous freezing conditions. The two immersion mode instruments compare well within uncertainties in the investigated temperature range and with respect to their instrument specific homogeneous freezing conditions (Fig. 2A). Kaolinite from the same supplier has also been used in previous work, e.g., in studies with PIMCA-

PINC and IMCA-ZINC (Kohn et al., 2016; Lüönd et al., 2010; Welti et al., 2012) and LACIS (Hartmann et al., 2016). Kohn et al. (2016) reported for $400\,\mathrm{nm}$ particles a $T_{50}$ of $238\,\mathrm{K}$, which agreed well with IMCA-ZINC experiments by Welti et al. (2012) when taking time dependence into account ($T_{50} = 238.5\,\mathrm{K}$ also for $400\,\mathrm{nm}$). The freezing curve, i.e. $T_{50}$ of $500\,\mathrm{nm}$ kaolinite particles measured with PIMCA-PINC in this study is about $3\,\mathrm{K}$ lower than the $400\,\mathrm{nm}$ particles used by Kohn et al. (2016) using the same instrument. Alternatively this can be viewed as for kaolinite, at fixed $T$ of $238\,\mathrm{K}$, the $FF$ is $\sim 30\,\%$

lower for $500\,\mathrm{nm}$ than for $400\,\mathrm{nm}$ particles previously investigated. This is surprising given the larger sized particles should





be more effective INPs. SBM fit lines (Fig. 2B/C) are based on data from Hartmann et al. (2016) investigating the same kaolinite sample, however, the SBM underestimates the $FF$ in the present study by up to $10 - 15\%$ in PIMCA-PINC and up to $10\%$ in LACIS throughout the investigated temperature range, although both PIMCA-PINC and LACIS results agree within measurement uncertainty to the fit curves (Fig. 2B/C). The fraction of multiple-charged particles is already considered in the

presented SBM fit curves in Fig. 2, thus not the reason for the discrepancy. A reason which may contribute to the discrepancy in ice activity when comparing to literature can be the method of particle generation such as dispersal from an aqueous solution or dry dispersal from a powder may influence the results. Particles in this work were generated from an aqueous suspension while Kohn et al. (2016), Welti et al. (2012), Lüönd et al. (2010) and Hartmann et al. (2016) examined dry dispersed particles. Producing particles from an aqueous suspension may e.g., lead to a redistribution of soluble material, effecting the exposure of

ice active sites on the particle surface. However in immersion mode a redistribution of solutes on the surface should not play a significant role since the solute should re-mobilize in the comparatively large droplets formed previous to ice nucleation. A decrease in the freezing temperature was previously observed for microcline during immersion freezing as mentioned above and also for Arizona test dust in water subsaturated conditions (Koehler et al., 2010). If at all, only a small difference within experimental uncertainty was found between Hartmann et al. (2016, dry dispersed particles) and LACIS in this study (wet

dispersed particles). This can be seen by comparing LACIS kaolinite measurements and respective SBM curves in Fig. 2C. Note, measurements with PIMCA-PINC using the same particles show a significant reduction in the ice activity with a $T_{50}$ of $\sim 2\,\mathrm{K}$ lower for wet generated kaolinite (Fluka) particles for measurements conducted at ETH Zurich in succession to LINC (see Appendix C for more details). This indicates that there is a change in the ice activity of kaolinite particles when suspended in water and that multiple-charged particles are not the sole reason for this discrepancy. We suggest that setup specific discrep-

ancies such as the method of particle generation and the quality of size-selection plays an non-negligible role which requires close attention to quantitatively compare INP measurements.

The broadest temperature range ($233 - 258\,\mathrm{K}$) investigated during LINC was for the two birch pollen washing waters (Fig. 2, lower panels; cyan and green). Parallel measurements with LACIS and PIMCA-PINC (Fig. 2A) agree well within their uncertainty. The birch pollen washing waters birchS and birchN are active below $258\,\mathrm{K}$. In comparison to the mineral dusts, the $FF$

shows a weaker temperature dependence and the $FF$ levels off below $\sim 248\,\mathrm{K}$, i.e. a fraction of more than $35\%$ of the droplets froze homogeneously. In the present study, the birchN sample ($300\,\mathrm{nm}$) shows a higher ice activity than birchS ($500\,\mathrm{nm}$). The results for the birchN sample compare well to Augustin-Bauditz et al. (2016), who tested the same sample batch (Fig. 2B/C) when comparing SBM curves. Using fit parameters for birchS from Augustin et al. (2013) the SBM curve overpredicts the $FF$ measured with both PIMCA-PINC and LACIS. The lower ice activity of birchS could arise from storage of the sample at room

temperature for more than three years between measurements done by Augustin et al. (2013) from which the SBM parameters were taken and measurements presented herein. The birchS sample seems to loose ice activity during storage suggesting that care should be taken when comparing results on ice nucleation of biological samples because some samples could potentially change over time. A loss of ice activity over time also has been previously observed for *Pseudomonas syringae* (e.g., Polen et al., 2016).





In order to quantify the overall instrumental differences between PIMCA-PINC and LACIS, the $FF$ data is averaged in 1 K bins and plotted in Fig. 3A. A good agreement is found between the two instruments within the experimental uncertainty for the majority of the data. By using the Bland-Altman approach (Bland and Altman, 1999), the difference in $FF$ between PIMCA-PINC and LACIS is calculated as a function of their obtained mean $FF$ and shows no trend (see Fig. 3B), implying

5   that no instrument specific offset in $FF$ is observed. This result suggests, that discrepancies found previously are not due to the performance of PIMCA-PINC or LACIS to accurately quantify immersion freezing as results are well within the measurement uncertainties of the instruments when operated in parallel. Instead, it suggests that differences in sample material or treatment of the particles prior to measurements such as the particle generation and the size-selection procedure, as discussed above could be the cause of previous discrepancies.

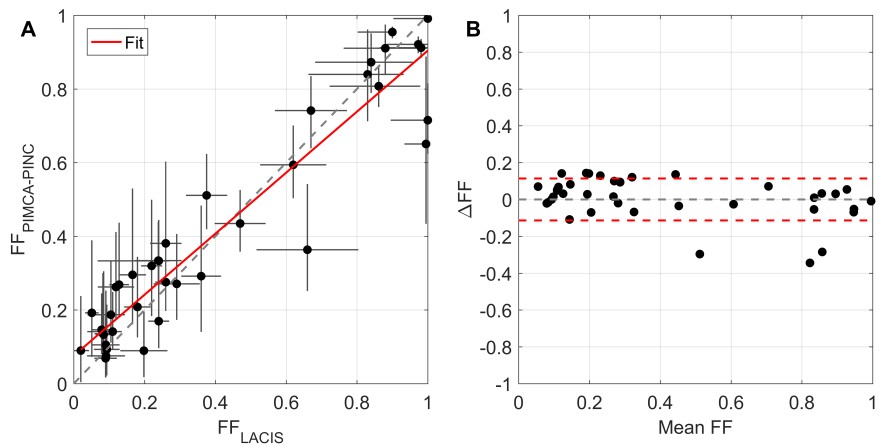

**Figure 3.** A) Correlation of frozen fraction for all aerosol types measured between PIMCA-PINC and LACIS. Data is binned in 1 K intervals for the correlation. The dashed line represents the 1:1-line and the red line is a linear fit to all samples. B) Differences between PIMCA-PINC and LACIS are shown as a function of their mean $FF$. Red dashed lines show the standard deviation ($1\sigma$) range.

### 3.2   Results of deposition and condensation mode measurements

For the first time PINC and SPIN measurements were conducted using the same sample and size-segregated aerosol particles. Experiments between 233 K and 253 K were performed with both instruments by scanning $RH$ from ice saturation up to above water saturation until droplet breakthrough was observed. The tested samples were: untreated and nitric acid treated microcline,

15   kaolinite and birch pollen washing water (birchN). The ice activity of the individual aerosol types is discussed on the basis of PINC measurements shown in Fig. 4 and the comparison between PINC and SPIN is discussed thereafter.

An active INP in the deposition mode is expected to have a high $AF$ or activate at lower $RH$ compared to less active INPs. PINC data indicate that the most active particles are not surprisingly found to be at 233 K with onset of ice formation at $RH_{\mathrm{w}}$ of $82 - 86\,\%$ corresponding to $RH_{\mathrm{i}} = 121 - 127\,\%$. Among the tested mineral dust samples, the untreated microcline particles





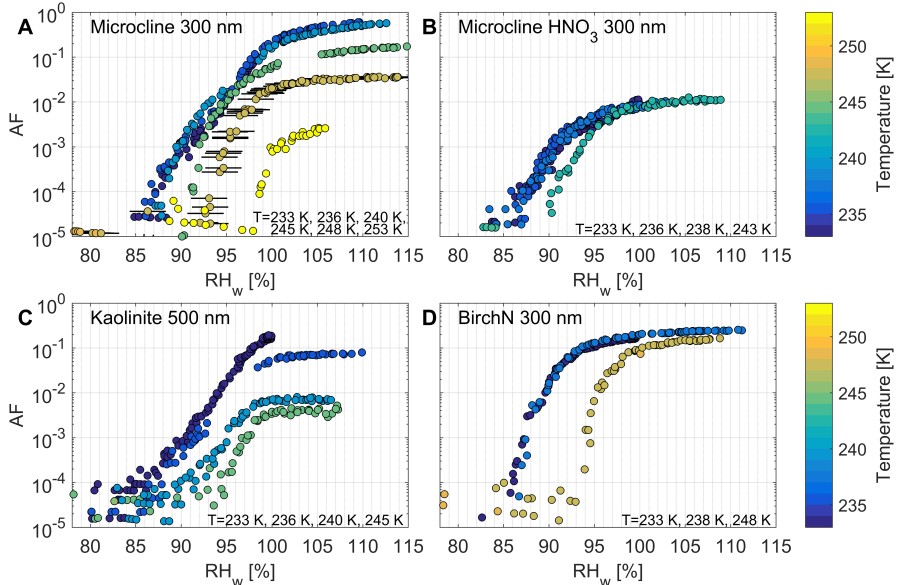

**Figure 4.** Activated fractions measured with PINC as a function of $RH_w$ at measured temperatures for four aerosol types: A) microcline untreated (300 nm) and B) microcline after treatment with $HNO_3$ (300 nm), C) kaolinite (500 nm) and D) birch pollen washing water (birchN, 300 nm). Exemplary measurement uncertainties are given for microcline (A) at 248 K.

(300 nm) show the highest $AF$ for a given $RH$ (Fig. 4A). The ice formation onset (here defined for convenience of discussion as $AF = 10^{-3}$) for untreated microcline is observed at $RH_w = 89\%$ ($RH_i = 120\%$) and the maximum $AF$ ranges from $6 \cdot 10^{-1}$ to $3 \cdot 10^{-3}$ at $236 - 253$ K respectively. Earlier deposition nucleation studies on K-feldspars observed a similar range of $AF$ (Yakobi-Hancock et al., 2013; Zimmermann et al., 2008). Treatment of microcline with nitric acid (Fig. 4B) resulted in

a lower maximum $AF$ of $10^{-2}$ for the temperature range investigated ($233 - 243$ K) compared to untreated microcline which showed $AF$ on the order of $10^{-1}$ for the same temperature range. The decrease in ice activity for acid treated particles was also observed in the immersion freezing experiments as discussed above. Freezing onset conditions do not significantly change with acid treatment below 240 K (Fig. 4B), but at 243 K a higher $RH_w$ is needed to reach $AF = 10^{-3}$. Kulkarni et al. (2014) also reported a reduction in the ice nucleation ability of 200 nm particles of a K-feldspar sample after coating with sulfuric

acid. A general reduction in the ice nucleation ability agrees with the immersion freezing measurements on microcline after treatment with nitric acid presented in this work, but sulfuric acid treatment was not tested with PINC. For a discussion of possible causes of the reduced ice activity after acid treatment we refer the reader to Sec. 3.1.

The ice nucleation activity of 500 nm kaolinite particles (Fig. 4C) was tested at four temperatures. Onset freezing conditions ($AF = 10^{-3}$) are observed at $RH_w = 90 - 98\%$. For temperatures 236 K, 240 K and 245 K a plateauing effect is observed for

$RH_w < 100\%$ indicating a saturation effect of ice nucleation occurring. It is unclear if this would also be observed at 233 K because the experiment was stopped shortly after reaching water saturation as the limit of the supersaturation attainable by



PINC (limited by the cooling power of the walls) was reached. When comparing to previous studies, a $15-20\%$ higher $RH_{\mathrm{w}}$ is required in this study to reach an $AF$ of $10^{-3}$ compared to Wex et al. (2014) at $T = 239-243\,\mathrm{K}$ using $300-700\,\mathrm{nm}$ particles. Also comparing to Welti et al. (2014), the required $RH_{\mathrm{w}}$ of $94\%$ ($T = 233\,\mathrm{K}$) to reach $AF = 10^{-2}$ in the current work is again $\sim 20\%$ higher (Welti et al., 2014, $RH_{\mathrm{w}} = 74\%$ at $T = 233\,\mathrm{K}$ using $400\,\mathrm{nm}$ particles). This indicates that kaolinite particles

investigated during LINC were less active INPs compared to previous studies. As mentioned before, a difference between previous studies and the present work is that aerosol particles in Welti et al. (2014) and Wex et al. (2014) were dry dispersed. In contrast to immersion freezing, wet dispersed particles can show a reduced deposition nucleation ability because of soluble material from an aqueous suspension having re-mobilized to the most hydrophilic locations on the particle surface during drying of the particles before being sampled into an INP counter. In such a case presumably the active sites on the particle are

blocked and it would be necessary for the $RH$ to overcome the deliquescence $RH$ of the soluble material to induce a phase change and further increase in $RH$ to overcome the solute effect followed by ice nucleation resulting in an observed delay in onset $RH$ of ice formation. These processes have already been suggested by earlier studies (e.g., Sullivan et al., 2010; Koehler et al., 2010; Alpert et al., 2011; Welti et al., 2014; Wex et al., 2014). Thus, the reduction in activity observed in the deposition nucleation regime suggests, but is not limited to an effect of the wet dispersion.

Birch pollen washing water (birchN sample, Fig. 4D) shows a steep increase and high maximum value in $AF$, suggesting uniformity among the particles responsible for ice nucleation. At $248\,\mathrm{K}$, ice nucleation onset occurs at $94\%$ $RH_{\mathrm{w}}$ and at $233\,\mathrm{K}$, the onset $RH_{\mathrm{w}}$ is $85\%$. There is only a small temperature dependence of the maximum $AF$ value of the pollen sample, suggesting a threshold temperature required for deposition nucleation on birchN. An additional decrease in temperature results in only a marginal increase in the activated fraction as shown in Fig. 4D.

**Comparison between SPIN and PINC**

In Fig. 5 we show SPIN and PINC data for all aerosol types and temperatures investigated with both instruments. Similar dependencies of $AF$ on $RH_{\mathrm{w}}$ and temperature are observed. Quantitatively, SPIN detects higher $AF$s, with differences more pronounced at lower temperatures and $RH_{\mathrm{w}}$. For the birchN, the difference at low temperatures is less pronounced than for the mineral dusts, suggesting an aerosol specific feature. The birchN particles are the most hygroscopic of the samples examined

in this work. The ice activity of particles influences the time for ice growth within the residence time of the chamber. Less efficient INPs display a higher nucleation time dependence. Due to non-instantaneous nucleation upon entering the chamber they have a shorter available growth time.

The largest deviation between PINC and SPIN is observed for measurements on nitric acid treated microcline, which showed a lower $AF$ measured with SPIN compared to PINC at $T$ of $238\,\mathrm{K}$ and $243\,\mathrm{K}$. Note that measurements on nitric acid treated

microcline were performed on two different batches of nitric acid treated samples, i.e. PINC and SPIN did not measure in parallel for this aerosol type and it is possible that the ice active material may not have been as thoroughly deactivated compared to the batch measured with SPIN.

Grouping the data in $\pm 2\%$ $RH_{\mathrm{w}}$ and $1\,\mathrm{K}$ temperature bins, $AF$s measured with PINC and SPIN can be compared (Fig. 6). For the lowest $AF$s close to the detection limits of SPIN and PINC, scattering is larger as can be seen by the differences





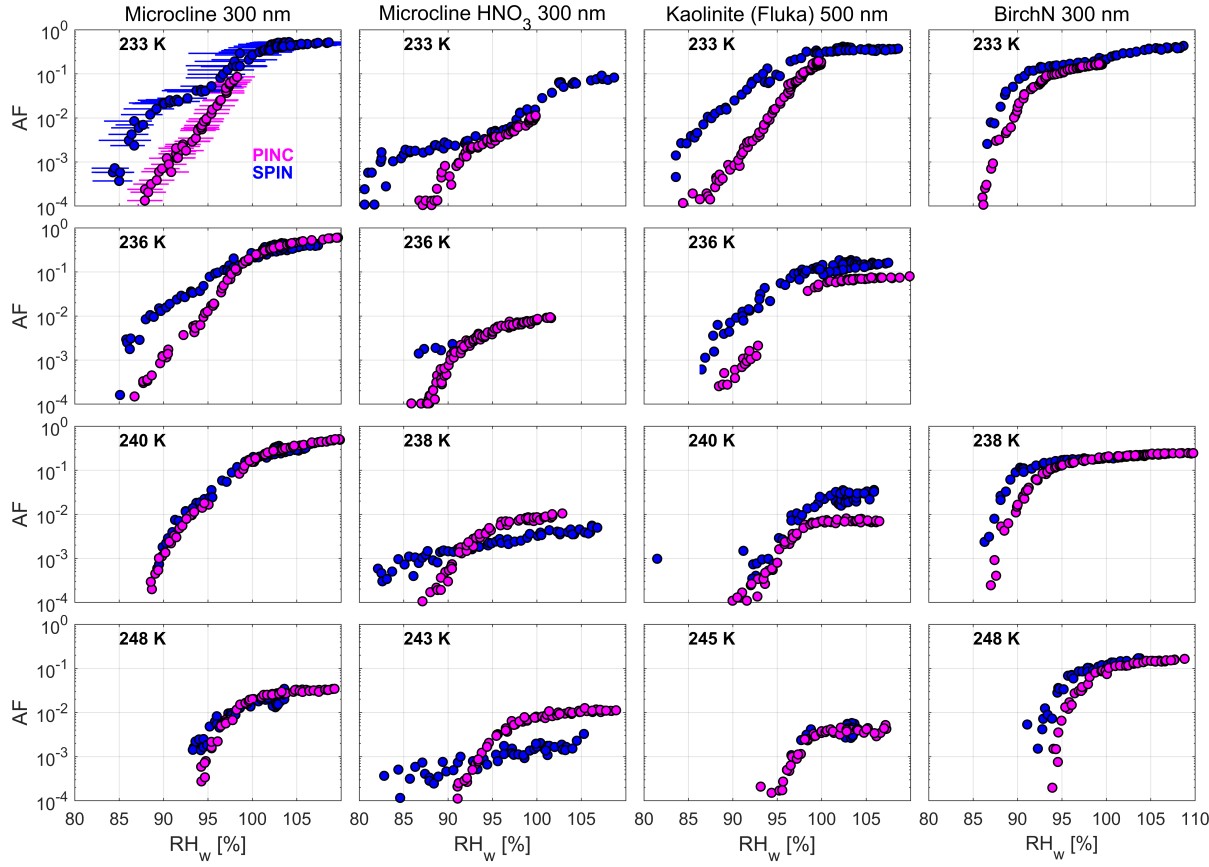

**Figure 5.** Resulting $AF$ from $RH_w$ scans measured with PINC (magenta) and SPIN (blue) for four aerosol types. The upper left panel includes an example of measurement uncertainties in $AF$ (not visible) and $RH_w$.

between SPIN and PINC as a function of the mean $AF$ (Fig. 6B). The deviation from the 1:1-line can be attributed to $T$ and $RH$ uncertainties, different data analysis procedures e.g. ice crystal threshold size and instrument design differences such as residence time. These differences are discussed in the following section.

As described in Sec. 2.3.1 and 2.3.3, chamber backgrounds are treated differently for the ice nucleation counters. For PINC,

5 the background is found to increase during the $RH$ scan and a typical background concentration of particle free air between the start and end of an $RH$ scan was $3.0 \, 1^{-1}$ obtained with a typical sample ice concentration of $6130 \, 1^{-1}$ on average for example for an experiment on kaolinite at $248 \, \mathrm{K}$. Typically the ice crystal concentrations in the experiment that reached $AF \geq 10^{-3}$ were sufficiently large that the background counts only played a minor role. The relative contribution of the background is higher at low $AF$ and $RH$. It is reasonable to assume that the background counts in SPIN also increase with increase in

10 $\Delta T$ (and $RH$) as is the case in PINC. However, the lack of a high $RH$ background measurement can yield a lower average background correction as a function of experimental time resulting in higher ice crystal counts for SPIN than PINC at the end of




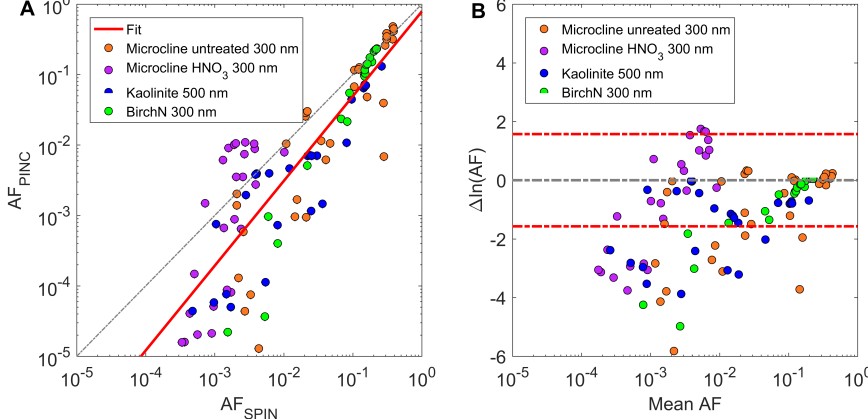

**Figure 6.** A) Correlation of $AF$ measured with PINC and SPIN differentiated for aerosol types. Data is binned in $\pm 2\%$ in $RH_{\mathrm{w}}$ to obtain the correlation. B) Differences in $AF$ between PINC and SPIN as a function of their mean $AF$ measured. The red dashed lines show the standard deviation ($1\sigma$) range.

a $RH$ scan. Note that background estimates for SPIN are justified by the fact that the contribution of aerosol particles is larger than that of the background counts arising from an increase in $RH$ and $\Delta T$. Even though background counts were estimated differently in the two experiments, the resulting change in $AF$ with or without background correction (see Appendix D for an example on PINC data) neither explains the discrepancies in $AF$ at high $RH_{\mathrm{w}}$ nor the difference in onset conditions. The

evaluation of the background could however still contribute to differences in observed $AF$ at low $RH_{\mathrm{w}}$ as shown by the yellow circles and crosses in Fig. D1 (Appendix D).

Another reason for observed differences between SPIN and PINC arises from the fact that ice crystals are identified using an experiment-specific size threshold to distinguish ice crystals from unactivated aerosol particles, which can complicate a direct comparison, especially at low $T$ and $RH$ where ice growth is kinetically limited. The comparison between PINC and SPIN was

performed with ice crystal size thresholds of $2$ and $2.5\,\mu\mathrm{m}$ respectively. The size thresholds were chosen such that ice crystals could accurately be counted while preventing unactivated particles being falsely counted as ice crystals. To demonstrate the effect of a change in the threshold size, Fig. 7 shows a comparison of the example of $RH$ scans on kaolinite using a $2\,\mu\mathrm{m}$ ice crystal threshold for PINC and either $2.5\,\mu\mathrm{m}$ or a $4\,\mu\mathrm{m}$ size threshold for SPIN. While the maximum $AF$ observed in SPIN did not change significantly with a change in the threshold size from $2.5\,\mu\mathrm{m}$ to $4\,\mu\mathrm{m}$, the freezing onsets ($AF = 10^{-3}$) increased

by $3 - 4\%$ $RH_{\mathrm{w}}$. Increasing the ice threshold to $4\,\mu\mathrm{m}$ in SPIN gives a better agreement to PINC onset conditions, but not in the maximum $AF$. Thus, changing the threshold size does not overcome the discrepancy in $AF$ observed with PINC and SPIN for kaolinite, which suggests that other factors such as time dependence of ice nucleation may contribute to the discrepancy for which the difference in the residence time in the chamber between SPIN ($t_{\mathrm{res}} \approx 9\,\mathrm{s}$) and PINC ($t_{\mathrm{res}} \approx 5\,\mathrm{s}$) plays a role. For aerosols that demonstrate a nucleation time dependence as has been shown with this kaolinite sample (Welti et al., 2012),

longer residence time allows more particles to act as INP and grow to larger ice crystal sizes before detection. In particular at





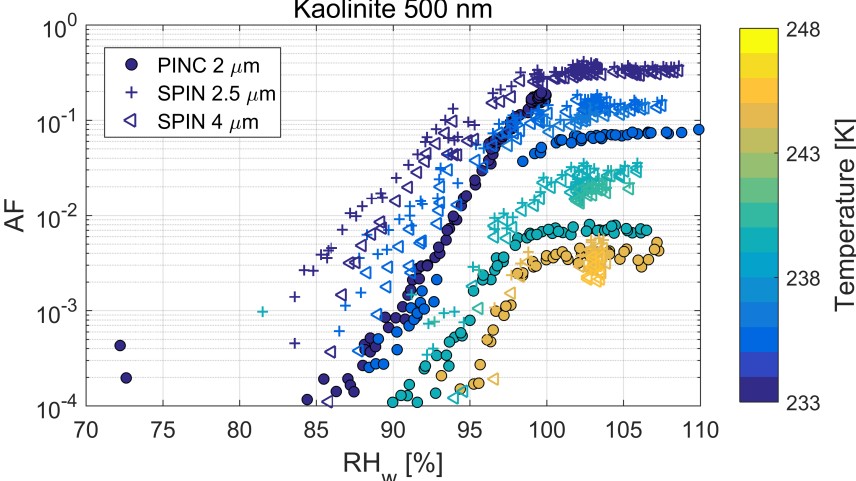

**Figure 7.** PINC and SPIN $RH$ scans for size-selected kaolinite particles. PINC data is analyzed with an ice crystal size threshold of $2\,\mu$m. SPIN data is presented with the used ice threshold size of $2.5\,\mu$m and a larger one of $4\,\mu$m.

low $T$, where the growth rates are lower, this could explain a higher $AF$ measured with SPIN compared to PINC. Following Rogers and Yau (1989) the crystal growth by diffusion for spherical ice crystals as a function of temperature was calculated for the typical residence times in PINC and SPIN (Fig. 8). The mass accommodation coefficient was set to $0.3$ based on literature data in Rogers and Yau (1989) ($0.2$ for small ice at $T > 263\,$K) and Skrotzki et al. (2013) ($0.2 - 1$ for $T = 190 - 235\,$K). The

initial starting particle diameter was set to $500\,$nm, the same diameter as used for kaolinite experiments. Note, that $t_{\text{res}}$ changes by $1 - 3\,$s depending on the experimental temperature and supersaturation. Assuming instantaneous nucleation of ice upon exposure of the aerosol particles to the chamber conditions, the growth calculations show that for a threshold size of $2\,\mu$m at $233\,$K, PINC would detect an ice crystal at $RH_{\text{w}} = 78.5\,\%$ and SPIN at $RH_{\text{w}} = 74\,\%$ (solid black lines/symbols, Fig. 8). The ice threshold size of $2.5\,\mu$m used for SPIN in this study accounts for the growth time effect (grey lines), which reduced the

observed difference in ice onset to $\sim 1\,\% \, RH_{\text{w}}$ between PINC and SPIN. While this resulting difference is small, note, that due to chamber flow dynamics, the particles are exposed to a $RH$ range across the aerosol sample lamina of $\pm 2\,\%$, depending on the nominal $T$ and $RH$ condition. Therefore, we expect a range of ice crystal sizes because of the range in $RH$. Further, the calculation shown in Fig. 8 assumes spherical ice crystal growth and also that nucleation is instantaneous and the entire residence time in the nucleation chamber is available for growth. If the latter two were not the case it would result in a reduction

of available time for ice crystal growth, and therefore larger differences would be expected at the position of detection for the two chambers. More efficient INPs would rapidly grow to ice crystals without a large time delay and support the hypothesis of ice growth effects and a weaker time dependence as can be observed for microcline at lower temperatures and birchN (Fig. 5). Instead, less efficient aerosol particles such as kaolinite with a demonstrated time dependence would have a smaller proportion of the residence time available for growth in PINC (residence time of $5\,$s) than in SPIN (residence time of $9\,$s). This time effect

could explain the offset between PINC and SPIN observed at a given $RH$.





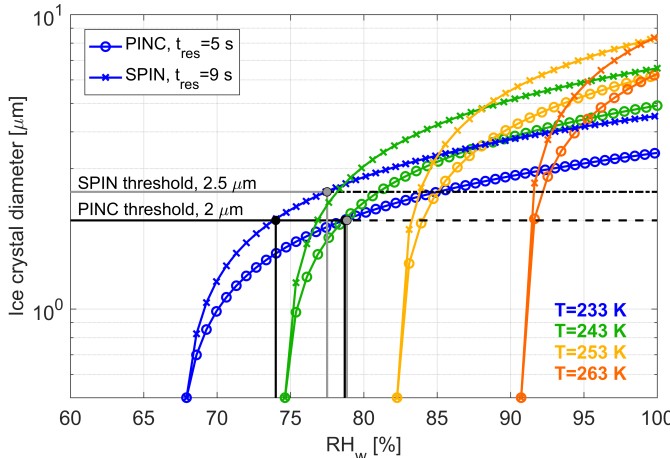

**Figure 8.** Ice crystal growth calculations for the typical residence times of $5\,\mathrm{s}$ in PINC (circles) and $9\,\mathrm{s}$ in SPIN (crosses), according to Rogers and Yau (1989) using a mass accommodation coefficient of 0.3. Vertical black lines show the discrepancy in $RH_\mathrm{w}$ arising from ice crystals counted with the same ice threshold size of $2\,\mu\mathrm{m}$ in both SPIN and PINC. Grey lines indicate the ice threshold size used during LINC resulting in a much smaller discrepancy in onset $RH_\mathrm{w}$. The used ice threshold sizes for PINC and SPIN are indicated by the horizontal dotted lines. Markers on the line plots are spaced by $1\,\%\,RH_\mathrm{i}$.

## 3.3 Comparing immersion and condensation freezing

It has been suggested that immersion mode is the dominant heterogeneous freezing pathway under mixed-phase cloud conditions (e.g., Ansmann et al., 2008). Recently, CFDCs have often been used for field measurements of INP concentration at water supersaturated conditions to represent immersion freezing (see Kanji et al., 2017, and references therein). As water supersat-
urated conditions in CFDCs should result in droplet formation followed by freezing at a constant temperature, CFDCs should simulate condensation freezing (see e.g., Welti et al., 2014, for a discussion of possible condensation freezing mechanisms). If condensation freezing in CFDCs is mechanistically different from immersion freezing, is doubtful as both nucleation mechanisms should proceed via the liquid phase with the requirement of overcoming an activation barrier of ice germ formation from liquid water molecules. How well CFDCs at or above water saturation compare with instruments that explicitly observe
immersion freezing has been addressed before (e.g., DeMott et al., 2015; Hiranuma et al., 2015; Garimella et al., 2017).
Here we compare measurements from PIMCA-PINC and LACIS to those from SPIN and PINC (Fig. 9). PIMCA-PINC and PINC cannot be operated at the same time and therefore experiments were repeated on different days. For the three tested aerosol types (microcline, kaolinite and birchN), a clear offset is found between measurements with PINC and SPIN compared to the immersion freezing experiments in PIMCA-PINC and LACIS (Fig. 9). Maximum $AF$ in PINC and SPIN does not exceed
$AF = 0.6$, even at $RH_\mathrm{w} > 105\,\%$ where droplet breakthrough biases the results. An $AF$ of approx. 0.6 was also the highest value reported for SPIN measurements in homogeneous freezing experiments, even well above water saturation and below $233\,\mathrm{K}$ in Garimella et al. (2016). Chamber characterization experiments with PINC revealed particle losses below $5\,\%$ (Boose



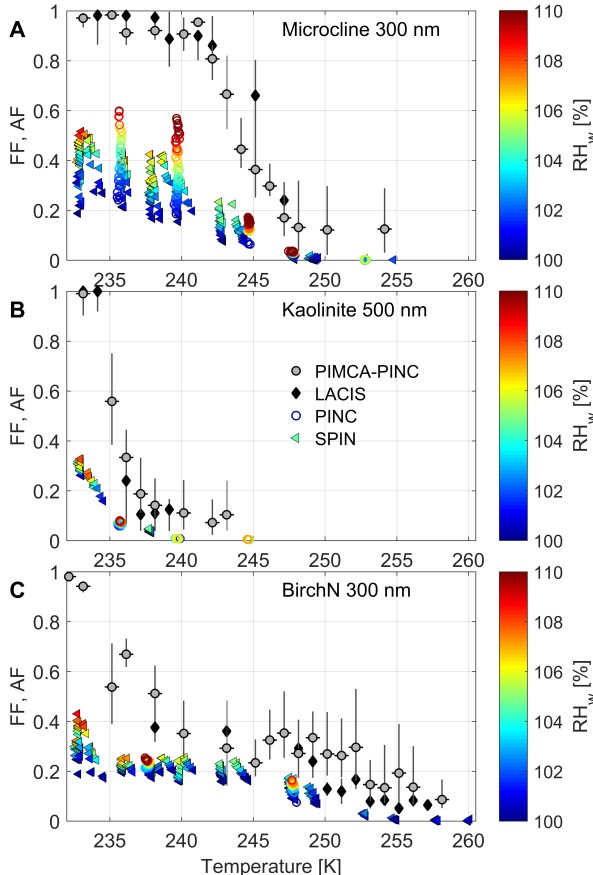

**Figure 9.** Comparison of all instruments. PIMCA-PINC and LACIS experiments were performed with droplet activation prior to exposure to freezing conditions. Results from immersion freezing experiments are reported as $FF$. PINC and SPIN measured the activated fraction ($AF$) at ice nucleation conditions above water saturation and $RH_w$ up to droplet breakthrough, which is the limitation for the scan range. $RH_w$ is indicated by the color bar. The uncertainty in $AF$ for PINC and SPIN is $14\%$.

et al., 2016) thus do not explain the observed difference between the CFDC (PINC) and immersion freezing (PIMCA-PINC, LACIS). In Fig. 10 a scatter plot for $FF$ of PIMCA-PINC and the $AF$ of PINC and SPIN obtained at $RH_w = 105\%$ are shown with respective lines for ratios of 1:1, 1:2 and 1:3 between the samples. LACIS data is excluded from the figure for clarity and not necessary for this discussion due to the good agreement found with PIMCA-PINC (Sec. 3.1). For microcline $FF$'s measured

5  with PIMCA-PINC are a factor of $2 - 3$ higher below $243\,\mathrm{K}$ and more than a factor of three higher at higher temperatures. A similar behavior is observed for kaolinite with factors of three or larger required to achieve agreement with immersion freezing. A factor of three difference between isothermal CFDC measurements and immersion freezing experiments has previously been reported by DeMott et al. (2015) comparing experiments on mineral dust between CSU-CFDC and the AIDA cloud chamber. In the present study, an offset between $FF$ and $AF$ is observed for all samples and in particular for the low $T$ measurements





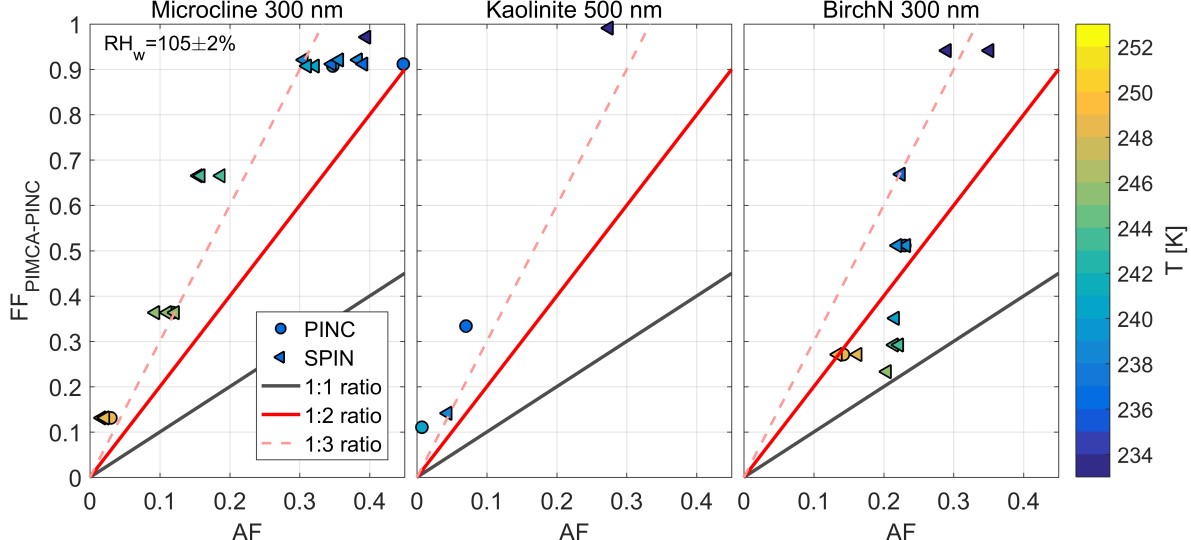

**Figure 10.** Scatter plot of $FF$ (immersion freezing) and $AF$ (condensation freezing) showing discrepancies between the instruments. Lines show the 1:1, 1:2, 1:3 ratios (black, red, pink respectively) for three different aerosol types. PINC and SPIN data are binned by $RH_\mathrm{w}$ of $\pm 2\%$.

the offset is not within measurement uncertainties. It is also noteworthy that the factor is found to change across the different experimental temperatures and aerosol types tested. As an example, for the birchN sample, the factor changed from less than two $T > 243\,\mathrm{K}$ up to even larger than three for $T \leqslant 235\,\mathrm{K}$.

The disagreement in ice activity observed with the two CFDCs and the two instruments explicitly measuring immersion freez-
ing provides evidence that CFDCs should not be assumed to give the same results as the existing in-situ experiments designed to exclusively measure immersion freezing. The results raise the question what possible differences can lead to this discrepancy and whether these are of physical or instrumental nature. If the ice nucleation mechanism is the same in all instruments (i.e. ice forms from liquid at the surface of an immersed particle), the most fundamental difference between condensation and immersion freezing experiments is the additional need to create liquid water during condensation freezing starting with a dry
aerosol particle or the presence of soluble material on the particle surface leading to freezing point depression. Recently, De-Mott et al. (2015) showed that aerosol particles can in fact be activated inside the CFDC to a sufficiently large droplet size to investigate immersion freezing in the CFDC for $RH_\mathrm{w}$ well in excess of $100\%$. However, at which time in the chamber droplet activation occurs and whether the residence time after droplet activation is sufficient to nucleate ice is unclear. Common ice nucleation counters operate on different residence times and if the time is not sufficient for droplet growth in the instrument
prior to freezing, a discrepancy is possible. The time lost nucleating and creating sufficient liquid within the growth section of the CFDC would cause a reduction in the observed condensation freezing, especially for INPs that show a time dependence for immersion freezing which is the case for the kaolinite sample used here (Welti et al., 2012).





In theory one could increase $RH_w$ until all particles are activated and nucleate ice, however the droplet breakthrough imposes limitations on the maximum attainable $RH$ in a CFDC. The question arises as to why such high $RH_w$ is required to see activation of these particles to ice crystals? It could be morphological or compositional heterogeneity even within a size-segregated sample implying that only a subset of particles are ice nucleation active. However, this is also observed with particles of uni-
form composition such as microcline. An increase in $\Delta T$ to increase $RH_w$ causes turbulence in the chamber and changes the flow dynamics in a CFDC at high $RH_w$ (Rogers, 1988; DeMott et al., 2015) especially for larger temperature gradients of about $10 - 15\,\mathrm{K}$ causing deviations from ideal flow conditions. Recently, Garimella et al. (2017) have provided further empirical evidence that aerosol particles may escape the lamina of CFDC type instruments resulting in particles being exposed to much lower $RH_w$ than predicted by ideal behavior of CFDCs and a resulting variable correction factor of $2.6 - 9.5$ was obtained from
their pulse tests. Particles escaping the lamina would require CFDCs to be operated at much higher $RH_w$ in order to activate all particles to water droplets to truly observe immersion freezing. Aerosol particles that escape the sample lamina cannot be expected to be processed at the set $T$ and $RH$ conditions (mean of the expected lamina) thus potentially leading to an underestimate of counted ice crystals. This underestimate may be more pronounced for higher $RH$ as the increase in turbulence may favor non-ideal conditions further supporting the differences observed between $FF$ and $AF$ in the work presented here.
Another difference between the CFDCs and the immersion mode instruments are the different methods of ice crystal detection. While the CFDCs detect the ice crystals as an absolute concentration (ratio of ice to total particle number entering the chamber) in the sample air volume via a size threshold, LACIS and PIMCA-PINC observe the relative fraction (ratio ice to total number of ice and droplets) via depolarization in a subset of the sample i.e., a relative count is used. The latter assumes that the sample volume is representative of the total sample air, with the advantage of being less sensitive to particle losses
in the chamber and counting efficiency errors arising from two different counting methods (OPC and CPC). In addition it is noteworthy that for LACIS, the optical detection instrumentation used here yielded $FF$s similar to an $AF$ from an absolute measurement method (Clauss et al., 2013) only when a correction factor normalizing to the particle concentration was applied as described in Niedermeier et al. (2010).





## 4    Summary and outlook

Experimental results of four online ice nucleation counters were compared using size-selected aerosol particles as INPs. Two devices designed to observe immersion freezing (PIMCA-PINC and LACIS) and two CFDCs for measuring deposition nucleation and condensation freezing (PINC and SPIN) were used in this study. The investigated aerosol samples were microcline

untreated and treated with either sulfuric or nitric acid, kaolinite (Fluka) and two types of birch pollen washing waters. The variety of samples allowed for measurements in the whole temperature (and $RH$) range possible with the chambers.

Treatment of the microcline sample with either sulfuric or nitric acid, followed by washing off the acid, destroyed the ice nucleation ability of the microcline permanently in immersion freezing for both sulfuric an nitric acid and led to a significantly reduced $AF$ in deposition nucleation and condensation freezing conditions between $233\,\mathrm{K}$ and $243\,\mathrm{K}$ when nitric acid treat-

ment was applied.

A comparison of parallel measurements with LACIS and PIMCA-PINC, conducted for the first time with these instruments, showed a very good agreement for the investigated aerosol types. No instrument specific differences for immersion freezing experiments were found in parallel measurements suggesting other factors such as the particle size-selection and dispersion method contributing to discrepancies found when comparing results from instruments operated at different times and places.

Measurements from the two CFDC instruments PINC and SPIN were compared in the sub- and supersaturated $RH_{\mathrm{w}}$ regime. Results showed qualitative agreement. However, a direct comparison showed that SPIN detects higher $AF$, in particular at low temperatures ($233 - 236\,\mathrm{K}$) and lower $RH_{\mathrm{w}}$. Calculations of ice crystal growth revealed that the chamber residence times, in addition to the selected ice crystal threshold sizes can largely explain these discrepancies and showed that their effects on reported results from a CFDC cannot be ignored.

Lastly, results from all four instruments were compared to investigate possible differences between condensation- ($RH_{\mathrm{w}} =$ $105\,\%$) and immersion freezing. Overall a clear discrepancy up to a factor of three or higher was found between immersion freezing and condensation freezing results, which is similar to the scaling factor of three as reported by DeMott et al. (2015) for mineral dust particles. This factor was observed to vary with aerosol type and temperature investigated in this work. When comparing CFDCs with chambers exclusively measuring immersion freezing, the detection methods used to evaluate $FF$ and

$AF$ should be kept in mind in addition to the $RH_{\mathrm{w}}$ of the CFDC at which the comparison is done. For instance, CFDC instruments report $AF$s by measuring absolute ice concentrations from an OPC, which are normalized to total sampled particles from a CPC, while immersion freezing $FF$s are obtained by normalizing using the ratio of ice crystals to the sum of ice and droplets from the same instrument. To determine the extent CFDCs are able to measure immersion freezing, further investigation at very high $RH_{\mathrm{w}}$ allowing for full droplet activation within the residence time of the chamber would be necessary.

An assessment if instruments measure the same physical mechanism (i.e. immersion freezing) cannot be made based on the present study. To which extent the observed deviations originate due to the different ice detection methods or residence time for droplet activation requires further investigation. For future studies the use of detectors measuring the absolute number of ice crystals and water droplets by depolarization would be advantageous.



## Appendix A: Soccer Ball Model (SBM)

The SBM can model temperature dependent frozen fractions for particles of different materials based on classical nucleation theory and was introduced in detail by Niedermeier et al. (2011). The ice nucleating sites of the material are represented by a contact angle distribution with an average contact angle, $\mu_\theta$, and the standard deviation, $\sigma_\theta$, together with an assumed size of the ice active sites, $S_{\text{site}}$. The abundance of these sites is given by $\lambda$, the average number of sites per particle. The nature of the ice nucleating sites of the respective material, is described by $\mu_\theta$ and $\sigma_\theta$ i.e., they represent a material property, while $\lambda$ represents the abundance of sites and might differ for different batches of the same material.

**Table A1.** Parameters used for SBM calculations shown in Fig. 2 with the fractions of multiple-charged particles (Table 1). The mean, $\mu_\theta$, and standard deviation, $\sigma_\theta$, of contact angle distribution, $\lambda$ as a function of the particle diameter ($D_\text{p}$) and $S_{\text{site}}$ are taken from literature.

| Aerosol type | $S_{\text{site}}$ [m$^2$] | $\mu_\theta$ [rad] | $\sigma_\theta$ [rad] | $\lambda$ | Reference |
|---|---|---|---|---|---|
| Microcline 200/300 nm | $10^{-14}$ | 1.29 | 0.10 | $2.03 \cdot 10^{13}\,\text{m}^{-2} \cdot D_\text{p}^2$ | Niedermeier et al. (2015) |
| Kaolinite 500 nm[a] | $10^{-14}$ | 1.87 | 0.25 | n.a. ($n_{\text{site}} = 3.14 \cdot 10^{12} \cdot D_\text{p}^2 + 0.0203$) | Hartmann et al. (2016) |
| BirchN 300 nm[b] | $3.14 \cdot 10^{-16}$ | 1.016 | 0.080 | (a) $3.30 \cdot 10^{12}\,\text{m}^{-2} \cdot D_\text{p}^2$ | Augustin-Bauditz et al. (2016) |
| | $3.14 \cdot 10^{-16}$ | 0.834 | 0.0005 | (b) $6.65 \cdot 10^{11}\,\text{m}^{-2} \cdot D_\text{p}^2$ | Augustin-Bauditz et al. (2016) |
| BirchS 500 nm | $3.14 \cdot 10^{-16}$ | 1.016 | 0.080 | $1.78 \cdot 10^{12}\,\text{m}^{-2} \cdot D_\text{p}^2$ | Augustin et al. (2013) |

[a] In the case of kaolinite, $n_{\text{site}}$ relates to a former version of the SBM and describes the number of surface sites which is assumed to be equal for equally sized particles.

[b] BirchN parameters are similar to those given in Augustin et al. (2013), with the exception of $\lambda$, i.e., the average number of ice active molecules per particle, which is different due to the use of a different batch of the birchN sample.

## Appendix B: Correcting the frozen fraction for multiple-charged particles

Previously, Hartmann et al. (2016) have introduced the correction for multiple-charged particles in the size distribution of quasi-monodisperse particles to inter-compare independent studies on INPs in the immersion mode.

According to Table 1, the $FF$ has been recalculated assuming that all particle sizes feature the identical heterogeneous nucleation rate ($J_{\text{het}}$):

$$FF_{\text{calc}} = a_1 \cdot (1 - \exp(J_{\text{het}} \cdot A_1 \cdot t_{\text{res}})) + a_2 \cdot (1 - \exp(J_{\text{het}} \cdot A_2 \cdot t_{\text{res}})) + a_3 \cdot (1 - \exp(J_{\text{het}} \cdot A_3 \cdot t_{\text{res}})) + ... \qquad (B1)$$

with $a_i$ being the fraction of particles with $i$ charges and the particle surface area of $A_i$. $J_{\text{het}}$ is chosen to reach the minimum squared error for the difference between $FF$ and $FF_{\text{calc}}$. The corrected frozen fraction $FF_{\text{corr}}$ is then obtained by

$$FF_{\text{corr}} = a_1 \cdot (1 - \exp(J_{\text{het}} \cdot A_1 \cdot t_{\text{res}})) \qquad (B2)$$

as shown in Fig. B1 for measurements with PIMCA-PINC and LACIS.





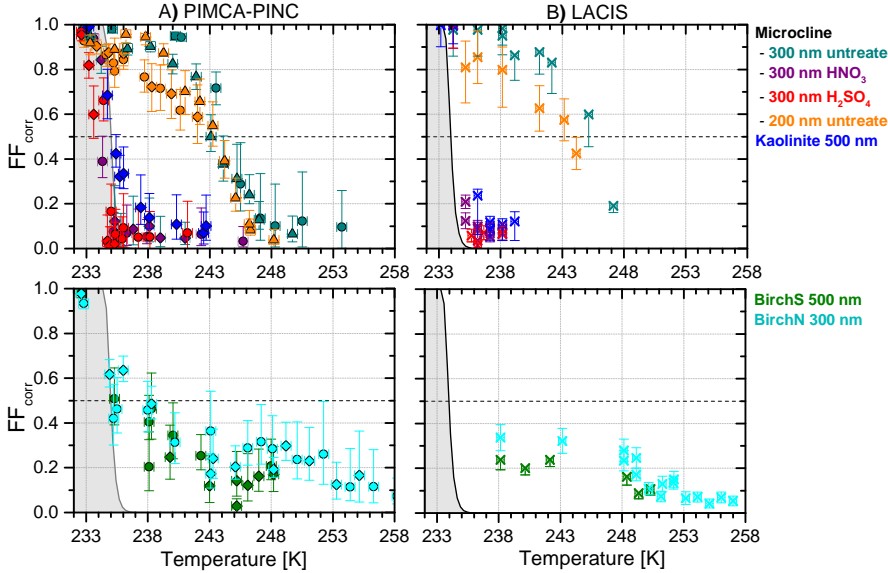

**Figure B1.** Frozen fraction for results obtained with PIMCA-PINC (A) and LACIS (B) accounting for the fraction of multiple-charged particles in the quasi mono-disperse sample given in Table 1.

## Appendix C: Immersion freezing of kaolinite with PIMCA-PINC using different particle generation methods

In succession of the LINC study, additional measurements were conducted with PIMCA-PINC using the same kaolinite sample. Results are shown in Fig. C1. Measurements with size-selected particles of $400\,\mathrm{nm}$ were compared when wet and dry generation methods were used. The particles were either dispersed from an aqueous solution via an atomizer similar to the method in this study or by dry dispersion using a Fluidized Bed Generator (TSI) as described in Kohn et al. (2016). Multiple individual measurements consisting of a temperature scan were conducted. Dry dispersed $FF$ measurements by Kohn et al. (2016) were reproducible. A difference in $FF$ based on the particle generation method is clearly observed in this comparison and in particular between temperatures of $235\,\mathrm{K}$ and $240\,\mathrm{K}$ found to exceed measurement uncertainty.





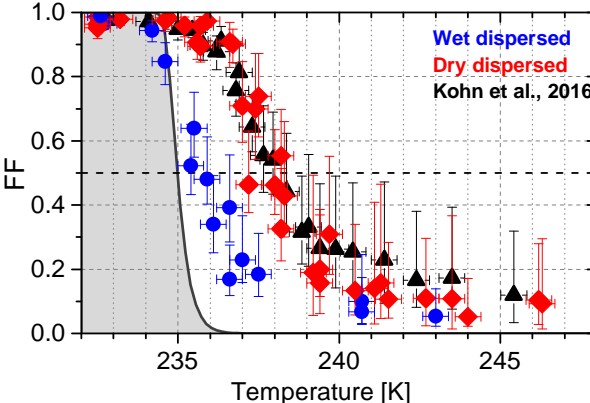

**Figure C1.** Frozen fraction for measurements of size-selected kaolinite (Fluka) particles (400 nm) using the PIMCA-PINC instrument after the LINC campaign (blue and red). Data from a previous study by Kohn et al. (2016) are shown for comparison (black; dry dispersed).

## Appendix D: Effect of instrument background correction on the activated fraction

Ice crystal counts in CFDCs are biased by background counts such as frost particles falling off the iced chambers walls, which are falsely counted as ice crystals. The background counts are evaluated for PINC at the beginning and at the end of each $RH$ scan. Linear interpolation between the two background measurements is used to determine background counts as a function of $RH$ which are then subtracted from the sample counts. The background correction for SPIN was conducted in a different manner in the presented inter-comparison study. Background counts are determined from the ice crystal counts at the start of $RH$ scans ($RH_i < 103\%$) where no ice nucleation is expected. Using sample air allows to include the concentration of false ice counts due to the measured aerosol population. The change in $AF$ by accounting for the background in a typical experiment during this study is shown on the example of kaolinite (Fig. D1). It shows that for high $AF$ (high $RH_w$) the correction has a minor effect (data points are overlapping). Thus, for measurements presented here the treatment of the background between SPIN and PINC does not affect the main findings. The interpolation through the scan for PINC would in particular effect the values at high $RH_w$. However, for low $AF$ closer to the freezing onset, the background correction reduces the $AF$. Thus, the method of accounting for the background does not explain differences observed between PINC and SPIN. Instead, it supports that for particle concentrations used in this study, accounting for the background counts does not have an influence on the results. It is noteworthy that a stronger effect may be found for experiments with low observed $AF$ or INP concentration e.g. when $RH$ scans are conducted in field studies for which the correction of the background has to be carefully considered.





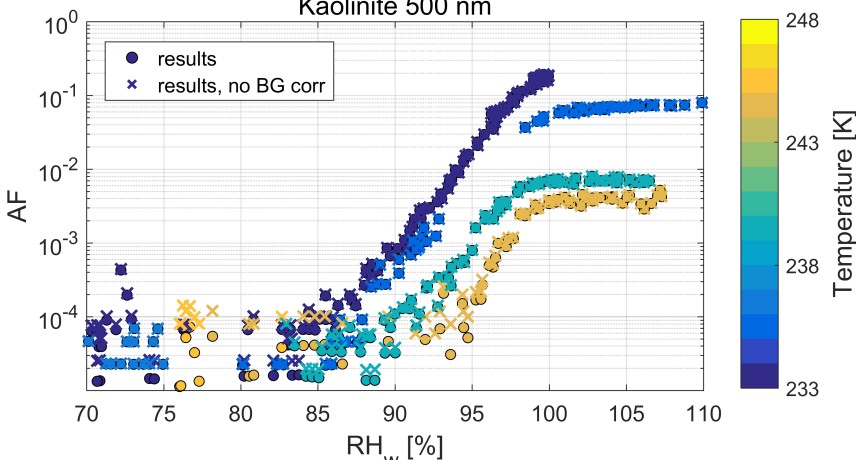

**Figure D1.** Activated fraction results of PINC including background correction (circles) as shown in Fig. 7 and raw data without accounting for background counts (crosses).

*Author contributions.* MBK prepared the manuscript with contributions from ZK, HW, SH, AW, SG and JA. MBK, ZK, AW and HW interpreted the data with contributions from SH and SG. MBK conducted and analyzed PIMCA-PINC and PINC measurements and prepared all overview figures of the manuscript; AW and PH run SPIN measurements and AW analyzed the data; HW oversaw and organized the campaign; SH, SG and LH operated LACIS and analyzed the data and all TROPOS participants helped with particle preparation and generation
5   as well as sizing.

*Acknowledgements.* M. Burkert-Kohn was funded by grant no. ETH-17 12-1, ETH Zurich. This work is partly funded by the German Research Foundation (DFG), Research Unit FOR 1525 (INUIT), project WE 4722/1-2 and has received funding from the European Union's Seventh Framework Programme (FP7/2007-2013) project BACCHUS under grant agreement no. 603445. We thank B. Würz from UFZ for giving support to use the centrifuge, Pharmallerga for providing pollen samples (birchS) and Hannes Wydler and Thomas Conrath for
10  technical support. The authors are grateful for editing of the manuscript by Robert O. David and M. Paramonov and for helpful comments on ice growth calculations from Paul J. DeMott.



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
