# Peer review of "Leipzig Ice Nucleation chamber Comparison (LINC): Inter-comparison of four online ice nucleation counters"

_Atmospheric Chemistry and Physics, 2017_

## Referee Comment (RC1) · Anonymous Referee #1 · 24 May 2017

Review of "Leipzig Ice Nucleation chamber Comparison (LINC): Inter-comparison of four online ice nucleation counters" by *Burkert-Kohn et al*.

**Summary and general comments**

This study presents a quantitative evaluation of the ice nucleation (IN) abilities of seven types of aerosol particles measured by four IN counters co-located at TROPOS, Leipzig. Based on the results obtained, the authors address the importance of the inter-comparison workshop with co-deployed instruments, uniform aerosol dispersion procedure and size segregation method. From my point of view, the difference in the ice crystal threshold sizes of PINC and SPIN is well justified (Sect. 3.2). Not employing upstream impactors to minimize biases of particle losses throughout this inter-comparison work was wise (P5L2-4).

Besides the suggestions for future studies made by the authors (P22L28-33), a comparative validation workshop of atmospherically representative ambient samples or in-situ field comparison of IN techniques (*DeMott et al*., 2017, ACPD) is also an important assignment for the IN research community. Finding a universal calibrant that can be used for validating any IN instruments at home bases should be kept as an alternative approach especially for those who may join this research field in the future.

I support publication of this manuscript after some minor comments below are properly addressed. Given the technical nature of the manuscript, it may be better published as a technical note in ACP or AMT than a regular research article according to the journal guidance:

www.atmospheric-chemistry-and-physics.net/about/manuscript_types.html.

I will leave this discussion to the authors and the editor.

**Minor/technical comments**

Section 3.3: It is not conclusive that the observed difference between *FF* and *AF* is due to different ice nucleation modes (that is, immersion vs. condensation) or technical artifacts/limitations (e.g., different methods in ice detection and IN efficiency estimation). Is it really fair to say what the authors present in this particular section is immersion vs. condensation? One can presume that the technical artifacts, such as ice detection, IN efficiency estimation (e.g., *FF* vs. *AF*), misalignment of particle stream in the chamber and inhomogeneous distribution of particles in individual droplets, play substantial role on potentially explaining the observed difference amongst the compared techniques. For this matter, the sub-title of this section should be named differently?

P1L10: Better read with "the whole range of atmospherically relevant thermodynamic conditions"

P2L19: The dominance of immersion freezing (P18 L2-3) may be better discussed in here. The extended discussion may be helpful to the reader.

P4L5-6: "…droplet containing a single aerosol particle…" - how good is this assumption? In reality, multiple particles might be in a droplet when aerosols were made using a suspension (*Emersic et al*., 2015, ACP; *Baydoun et al*., 2016, ACP; also your own statement in P9L4-5). This may be RH & droplet size dependent. Would this factor be important to interpret the difference between *FF* and *AF*?

P14 L12: *Sullivan et al*., 2010, GRL – authors perhaps meant to cite the following paper?

Sullivan, R. C. et al. (2010, ACP), Irreversible loss of ice nucleation active sites in mineral dust particles caused by sulphuric acid condensation, Atmos. Chem. Phys., 10 ,11,471–1 1,487, doi :10.5194/acp-10-11471-2010.

*Sullivan et al*. (2010, GRL) demonstrated that condensation/diffusion of hygroscopic materials could make particles ice active in immersion/condensation mode (which may be discussed anyway in the paper separately…).

P14L24-25: "The brichN particles are the most hygroscopic…" - based on what? CCNC? Either data or reference is missing.

P21L4-5: "…particles of uniform composition such as microcline" - I disagree with this statement. The authors state that their microcline sample contains bi-components (P4L10-12).

P22L7-10: I suggest separating into two sentences to improve the clarity of the statement - e.g., "Treatment of the microcline sample with either sulfuric or nitric acid…permanently in immersion freezing. In addition, the nitric acid treatment lead to…between 233 K and 243 K.".

---

## Referee Comment (RC2) · Anonymous Referee #2 · 26 May 2017

Review of "Leipzig Ice Nucleation chamber Comparison (LINC): Inter-comparison of four online ice nucleation counters" by Burkert-Kohn et al.

**General Comment:**

This manuscript presents the results of the direct inter-comparison of four cloud chambers (i.e., The Leipzig aerosol cloud interaction simulator (LACIS), the Portable immersion mode cooling chamber coupled to the portable ice nucleation chamber (PIMCA-PINC), the Portable ice nucleation chamber (PINC), and the spectrometer for ice nuclei (SPIN)). Different size-selected (200, 300, and 500 nm) uncoated (microcline, kaolinite, birch pollen) and coated (microcline/$H_2SO_4$, microcline/$HNO_3$) aerosol types were used to determine the performance of these INP counters. Given the versatility of the used cloud chamber, three out of the four heterogeneous ice nucleation modes were also studied (i.e., deposition nucleation, condensation and immersion freezing).

Although this manuscript does not present a completely novel idea given that this approach was introduced/presented in earlier studies, it is the first time that these four instruments are directly compared. This is of high value for the ice nucleation community, especially because the SPIN is a commercial instrument that will be used in several studies. There is an urgent need to understand what different cloud chamber measures, and this study provides very useful information that helps cloud chamber users to interpret their results. The manuscripts is very well written and the figures are easy to follow. The experiments were carefully designed and the results are accurately interpreted. This manuscript can be accepted in ACP after the following minor corrections are taken into account.

**Minor comments:**

1. I am wondering if the authors can comment about the following question. Out of the four instruments investigated in this study, what is the most reliable? Is there a "standard" instrument that can be used to validate a newly designed INP counter?
2. I like Figure 2 but it took me sometime to follow it. It may be more useful to use filled and open symbols here.
3. I did not understand why the LACIS modeled results by the SMB reached a plateau but not the PIMCA-PINC? Is it the model instrument specific?

**Specific comments:**

Page 2, line 3: Do the authors mean: "its importance"?

Page 2, line 3: I don't think this is the only paper showing this. Please add more references here.

Page 2, line 4: Spell-out INP.

Page 2, line 6: Add references after "climate".

Page 2, line 19: Replace "which" with "what".

Page 2, line 20: Add references after "understood".

Page 2, lines 20-22: I don't understand what the authors want to communicate here. Please clarify it.

Page 2, line 25: hourly? daily?

Page 2, line 33: DeMott et al., 2016 could be add it to the list.

Page 3, line 10: Please state what type of CFDC the authors refer to. CSU?

Page 3, line 12: Please state what type of mineral dust was used.

Page 3, line 33: Add references after "activity".

Page 4, line 23: I found the pore size quite big. Is there any chance that small fragments of the pollen grains could go through the filter pores?

Page 4, line 31 and Figure 1: I think it should be "Nuclei" instead of "Nucleus".

Page 5, line 12: "parallel" is out of place.

Page 6, line 12: Please state what diameter the authors are referring to.

Page 7, line 29-30: Based on Garimella et al. (2016), the SPIN has a depolarization optical detector. I am wondering why the authors did not use this to discriminate between ice particles and droplets, instead to focus on their size only?

Page 8, line 6: Delete Hartmann et al., 2011.

Page 10, line 25: I am wondering why the authors did not perform an experiment with 500 nm microcline particles as done for kaolinite and Birch pollen.

Page 11, line 1: Add references after "INPs".

Page 11, line 16: "note" is out of place.

Page 12, line 18-19: This is a bit unclear. Please clarify it.

Page 14, line 28-32: Given that these experiments differ from the others (i.e., they are not directly comparable), I am wondering if this should be removed for consistency.

Page 17, line 11-12 and Figure 7: Add "W" to RH.

Page 18, line 3: I don't think this is the only paper showing this. Please add more references here.

Table 1. Why is microcline twice here?

**References**

DeMott, P. J., Hill, T. C. J., McCluskey, C. S., Prather, K. A., Collins, D. B., Sullivan, R. C., Ruppel, M. J., Mason, R. H., Irish, V. E., Lee, T., Hwang, C. Y., Rhee, T. S., Snider, J. R., McMeeking, G. R., Dhaniyala, S., Lewis, E. R., Wentzell, J. J. B., Abbatt, J., Lee, C., Sultana, C. M., Ault, A. P., Axson, J. L., Martinez, M. D., Venero, I., Santos-Figueroa, G., Stokes, M. D., Deane, G. B., Mayol-Bracero, O. L., Grassian, V. H., Bertram, T. H., Bertram, A. K., Moffett, B. F., and Franc, G. D.: Sea spray aerosol as a unique source of ice nucleating particles, P. Natl. Acad. Sci. USA, 113, 5797–5803, doi:10.1073/pnas.1514034112, 2016

Garimella, S., Kristensen, T. B., Ignatius, K., Welti, A., Voigtländer, J., Kulkarni, G. R., Sagan, F., Kok, G. L., Dorsey, J., Nichman, L., Rothenberg, D. A., Rösch, M., Kirchgäßner, A. C. R., Ladkin, R., Wex, H., Wilson, T. W., Ladino, L. A., Abbatt, J. P. D., Stetzer, O., 15 Lohmann, U., Stratmann, F., and Cziczo, D. J.: The SPectrometer for Ice Nuclei (SPIN): an instrument to investigate ice nucleation, Atmos. Meas. Tech., 9, 2781–2795, doi:10.5194/amt-9-2781-2016, 2016.

---

## Author Comment (AC1) · 18 Aug 2017

**Response to comments from Referee #1**

We thank the Referee for their comments. Response is given in black and respective changes to the manuscript in *italics*. The Referee comments are reproduced in *blue*.
* * *
*Review of "Leipzig Ice Nucleation chamber Comparison (LINC): Inter-comparison of four online ice nucleation counters" by Burkert-Kohn et al.*

*Summary and general comments*
*This study presents a quantitative evaluation of the ice nucleation (IN) abilities of seven types of aerosol particles measured by four IN counters co-located at TROPOS, Leipzig. Based on the results obtained, the authors address the importance of the inter-comparison workshop with co-deployed instruments, uniform aerosol dispersion procedure and size segregation method. From my point of view, the difference in the ice crystal threshold sizes of PINC and SPIN is well justified (Sect. 3.2). Not employing upstream impactors to minimize biases of particle losses throughout this inter-comparison work was wise (P5L2-4).*
*Besides the suggestions for future studies made by the authors (P22L28-33), a comparative validation workshop of atmospherically representative ambient samples or in-situ field comparison of IN techniques (DeMott et al., 2017, ACPD) is also an important assignment for the IN research community. Finding a universal calibrant that can be used for validating any IN instruments at home bases should be kept as an alternative approach especially for those who may join this research field in the future.*
*I support publication of this manuscript after some minor comments below are properly addressed. Given the technical nature of the manuscript, it may be better published as a technical note in ACP or AMT than a regular research article according to the journal guidance:*
*www.atmospheric-chemistry-and-physics.net/about/manuscript_types.html.*
*I will leave this discussion to the authors and the editor.*

The authors thank the Referee for their comments on suggestions and provide responses to their comments and questions below:

We acknowledge the reviewer's comment to move this paper to AMT due to the technical nature of some parts. However, we suggest retaining the paper in ACPD/ACP because the instruments inter-compared in this paper are of a common design/identical to instruments already used to publish results in ACP and future publications with similar instruments (e.g. SPIN) are anticipated in ACP. So it would be unusual that a paper addressing comparisons of instruments whose results have been and will be published in ACP is itself published in AMT.
Secondly, the inter-comparison is based on results of non-technical aspects, which are
1. The investigation of the ice nucleation ability of birch pollen washing waters (biological) using continuous flow diffusion chambers for experiments in the deposition and condensation freezing regime
2. A study on nitric and sulfuric acid treated microcline for which the acid has been completely removed prior to the experiments. The results address conditions of immersion freezing (nitric/sulfuric acid treatment), as well as condensation freezing (nitric acid treatment).

*Minor/technical comments*
*Section 3.3: It is not conclusive that the observed difference between FF and AF is due to different ice nucleation modes (that is, immersion vs. condensation) or technical artifacts/limitations (e.g., different methods in ice detection and IN efficiency estimation).*
The following sentences have been added to the manuscript: *"Possible reasons for observed differences, such as technical artifacts or differences in the ice nucleation modes, are discussed. For simplicity, FF is used for experiments exclusively performed in the immersion mode. In contrast, for ice nucleation chambers measuring in the condensation mode, which is in this case not explicitly*

*distinguishable from immersion freezing, data is presented as AF in the following figures."* (p. 19, line 15 ff).

*Is it really fair to say what the authors present in this particular section is immersion vs. condensation? One can presume that the technical artifacts, such as ice detection, IN efficiency estimation (e.g., FF vs. AF), misalignment of particle stream in the chamber and inhomogeneous distribution of particles in individual droplets, play substantial role on potentially explaining the observed difference amongst the compared techniques. For this matter, the sub-title of this section should be named differently?*
We agree with the reviewer and have changed the name of section 3.3 to "*Apparent differences between immersion and condensation freezing*". (p. 19, line 5)

*P1L10: Better read with "the whole range of atmospherically relevant thermodynamic conditions"*
The suggested change has been made (p. 1, line 10).

*P2L19: The dominance of immersion freezing (P18 L2-3) may be better discussed in here. The extended discussion may be helpful to the reader.*
We accepted this valuable suggestion. The explanation (p. 18, line 2-3 in the initial manuscript) has been moved and implemented as follows: *"Further, Marcolli (2014) suggested that deposition nucleation might in fact be immersion freezing (or homogeneous freezing for T < 235 K) of water trapped in pores and cavities at water subsaturated conditions. Which ice nucleation pathways exist and under what conditions they are relevant in the atmosphere is not fully understood, but has been speculated and discussed (e.g., Kanji et al., 2017). It has been suggested that immersion mode is the dominant heterogeneous freezing pathway under mixed-phase cloud conditions (e.g., Ansmann et al., 2008, de Boer et al., 2011, Westbrook and Illingworth, 2011)."* (p. 2, line 19 ff)

*P4L5-6: "…droplet containing a single aerosol particle…" - how good is this assumption? In reality, multiple particles might be in a droplet when aerosols were made using a suspension (Emersic et al., 2015, ACP; Baydoun et al., 2016, ACP; also your own statement in P9L4-5). This may be RH & droplet size dependent. Would this factor be important to interpret the difference between FF and AF?*
Indeed particles from suspensions may contain agglomerates, particularly when comparably large droplets are generated directly from suspensions. These are the kind of droplets that are referred to in the literature you cite above. However, in our case, the suspension is atomized and the resulting droplets are then dried and the residual dry particles are size selected. Based on measurements done earlier in other experiments, we know from dispersion of dry particles from the mineral dust samples, that small enough particles in mineral dust samples are not available so as to form agglomerates amounting to the sizes studied in this work (200 – 500 nm). However, as this cannot be ruled out completely and to clarify this, we added the following: "*For insoluble materials such as kaolinite or feldspar, such a particle could consist of an agglomerate of smaller primary particles. However, the number concentration of primary particles in the dry sample strongly decreases with size for the mineral dust samples and the size range used in this study, which makes the presence of dust agglomerates unlikely.*" (p. 5, line 1 ff).

For the birch pollen samples it has been shown that the suspension contains small macromolecules, in which case agglomerates of molecules are produced. The following sentence has been added to the manuscript: *"For suspension of birch washing waters containing small macromolecules, an agglomerate of molecules is produced, which is referred to as a (single) aerosol particle after size selection in this work…".* (p. 5, line 4)

Ice nucleation instruments obtain AF and FF, which are based on particle counting after suspension, drying and size selection, as such single particles (or similar agglomerates of molecules for birch washing waters) can be assumed to be investigated in both cases. If they exist, any agglomeration should be systematically biasing the variables in FF and AF and therefore cannot be used to explain the

difference between AF and FF. We have reworded the mentioned sentence and it now reads *"…droplets containing single-immersed aerosol particles…"*. (p. 4, line 9-10)

*P14 L12: Sullivan et al., 2010, GRL – authors perhaps meant to cite the following paper?*

> *Sullivan, R. C. et al. (2010, ACP), Irreversible loss of ice nucleation active sites in mineral dust particles caused by sulphuric acid condensation, Atmos. Chem. Phys., 10 ,11,471–1 1,487, doi :10.5194/acp-10-11471-2010.*

*Sullivan et al. (2010, GRL) demonstrated that condensation/diffusion of hygroscopic materials could make particles ice active in immersion/condensation mode (which may be discussed anyway in the paper separately…).*

The reviewer correctly pointed out that the Sullivan et al., (2010) paper in ACP was to be cited here and was accidently cited as the GRL 2010 manuscript by the same first author. In the revised manuscript Sullivan et al. (2010, GRL) has been replaced by Sullivan et al. (2010, ACP). (p. 14, line 10).

*P14L24-25: "The brichN particles are the most hygroscopic…" - based on what? CCNC? Either data or reference is missing.*

Indeed, this statement is based on CCNC measurements, which were continuously conducted in parallel to the INP measurements. For birchN, already at SS = 0.1% (i.e., $RH_w$ = 100.1%, the lowest supersaturation sampled) all particles with a diameter of 300 nm were activated to droplets (CCN/CN = 1), while for all other samples, particle hygroscopicity could be derived. The information about the source of our statement was added to the text: *"The birchN particles are the most hygroscopic particles of the samples examined in this work, which was deduced from CCNC measurements where 300 nm particles fully activated at a supersaturation of 0.1% (i.e., the lowest supersaturation sampled), while for all other samples, particle hygroscopicity could be derived, i.e. 50% active fraction was achieved at a higher supersaturation."* (p. 15, line 7 ff).

*P21L4-5: "…particles of uniform composition such as microcline" - I disagree with this statement. The authors state that their microcline sample contains bi-components (P4L10-12).*

We agree with the reviewer and have deleted the respective sentence in the manuscript.

*P22L7-10: I suggest separating into two sentences to improve the clarity of the statement - e.g., "Treatment of the microcline sample with either sulfuric or nitric acid…permanently in immersion freezing. In addition, the nitric acid treatment lead to…between 233 K and 243 K.".*

The suggested change has been made and the section reads as follows: *"Treatment of the microcline sample with either sulfuric or nitric acid, followed by washing off the acid, destroyed the ice nucleation ability of the microcline permanently in immersion freezing mode. In addition, the nitric acid treatment led to a significantly reduced AF in deposition nucleation and condensation freezing conditions between 233 K and 243 K."* (p. 22, line 19 ff)

**References**

Ansmann, A., Tesche, M., Althausen, D., Müller, D., Seifert, P., Freudenthaler, V., Heese, B., Wiegner, M., Pisani, G., Knippertz, P., and Dubovik, O.: Influence of Saharan dust on cloud glaciation in southern Morocco during the Saharan Mineral Dust Experiment, J. Geophys. Res., 113, doi:10.1029/2007JD008785, d04210, 2008.

Beydoun, H., Polen, M., and Sullivan, R. C.: Effect of particle surface area on ice active site densities retrieved from droplet freezing spectra, Atmos. Chem. Phys., 16, 13359-13378, doi:10.5194/acp-16-13359-2016, 2016.

de Boer, G., Morrison, H., Shupe, M. D., and Hildner, R.: Evidence of liquid dependent ice nucleation in high-latitude stratiform clouds from surface remote sensors, Geophys. Res. Lett., 38, doi:10.1029/2010GL046016, 01803, 2011.

DeMott, P. J., Hill, T. C. J., Petters, M. D., Bertram, A. K., Tobo, Y., Mason, R. H., Suski, K. J., McCluskey, C. S., Levin, E. J. T., Schill, G. P., Boose, Y., Rauker, A. M., Miller, A. J., Zaragoza, J., Rocci, K., Rothfuss, N. E., Taylor, H. P., Hader, J. D., Chou, C., Huffman, J. A., Pöschl, U., Prenni, A. J., and Kreidenweis, S. M.: Comparative measurements of ambient atmospheric concentrations of ice nucleating particles using multiple

immersion freezing methods and a continuous flow diffusion chamber, Atmos. Chem. Phys. Discuss., 2017, 1–29, doi:10.5194/acp-2017-417, 2017.

Emersic, C., Connolly, P. J., Boult, S., Campana, M., and Li, Z.: Investigating the discrepancy between wet-suspension- and dry-dispersion-derived ice nucleation efficiency of mineral particles, Atmos. Chem. Phys., 15, 11311-11326, doi:10.5194/acp-15-11311-2015, 2015.

Kanji, Z. A., Ladino, L. A., Wex, H., Boose, Y., Burkert-Kohn, M., Cziczo, D. J., and Krämer, M.: Ice Formation and Evolution in Clouds and Precipitation: Measurement and Modeling Challenges, Chapter 1: Overview of Ice Nucleating Particles, Meteor. Monogr, doi:10.1175/AMSMONOGRAPHS-D-16-0006.1, 2017.

Sullivan, R. C., Petters, M. D., DeMott, P. J., Kreidenweis, S. M.,Wex, H., Niedermeier, D., Hartmann, S., Clauss, T., Stratmann, F., Reitz, P., Schneider, J., and Sierau, B.: Irreversible loss of ice nucleation active sites in mineral dust particles caused by sulphuric acid condensation, Atmos. Chem. Phys., 10, 11 471–11 487, doi:10.5194/acp-10-11471-2010, 2010.

Westbrook, C. D. and Illingworth, A. J.: Evidence that ice forms primarily in supercooled liquid clouds at temperatures >-27°C, Geophys. Res. Lett., 38, doi: 10.1029/2011GL048021, 2011.

---

## Author Comment (AC2) · 18 Aug 2017

Response to comments from Referee #2

Referee comments are reproduced in *blue*. Response is given in black and respective changes to the manuscript in *italics*.
* * *
*Review of "Leipzig Ice Nucleation chamber Comparison (LINC): Inter-comparison of four online ice nucleation counters" by Burkert-Kohn et al.*

*General Comment:*
*This manuscript presents the results of the direct inter-comparison of four cloud chambers (i.e., The Leipzig aerosol cloud interaction simulator (LACIS), the Portable immersion mode cooling chamber coupled to the portable ice nucleation chamber (PIMCA-PINC), the Portable ice nucleation chamber (PINC), and the spectrometer for ice nuclei (SPIN)). Different size-selected (200, 300, and 500 nm) uncoated (microcline, kaolinite, birch pollen) and coated (microcline/H2SO4, microcline/HNO3) aerosol types were used to determine the performance of these INP counters. Given the versatility of the used cloud chamber, three out of the four heterogeneous ice nucleation modes were also studied (i.e., deposition nucleation, condensation and immersion freezing).*
*Although this manuscript does not present a completely novel idea given that this approach was introduced/presented in earlier studies, it is the first time that these four instruments are directly compared. This is of high value for the ice nucleation community, especially because the SPIN is a commercial instrument that will be used in several studies. There is an urgent need to understand what different cloud chamber measures, and this study provides very useful information that helps cloud chamber users to interpret their results. The manuscripts is very well written and the figures are easy to follow. The experiments were carefully designed and the results are accurately interpreted. This manuscript can be accepted in ACP after the following minor corrections are taken into account.*

The Authors thank the Reviewer for the comments and address the minor comments below:

*Minor comments:*

1. *I am wondering if the authors can comment about the following question. Out of the four instruments investigated in this study, what is the most reliable? Is there a "standard" instrument that can be used to validate a newly designed INP counter?*

In this study, we address the inter-comparison of four ice nucleation chambers with their similarities and differences, including three chamber configurations that have been used in the field for sampling ambient INPs. We refrain from endorsing one particular chamber precisely because what we demonstrate in the manuscript is, that a number of factors need to be considered to interpret the data and inter-compare. Our goal here is to bring to attention to the community these different aspects and factors that need to be considered. Endorsing one INP counter also depends on what the research goals are for a given field or laboratory experiment. As has been pointed out recently in Cziczo et al. (2017) that a variety of instruments and methods would be necessary to cover the dynamic T and RH range and particle sizes for investigating heterogeneous ice nucleation in the atmosphere.

2. *I like Figure 2 but it took me sometime to follow it. It may be more useful to use filled and open symbols here.*
We agree with the reviewer. In the current version symbols only distinguish PIMCA-PINC experiments (circles) and LACIS (crosses) for clarity and different colors have been used for different aerosol samples investigated as in the previous version of the manuscript. We choose crosses for LACIS data rather than open symbols due to the number of (partly overlapping) data points in particular in Panel A (Fig. 2 and Fig. B1). We think the use of a different shape improves the readability of the plot.

*3. I did not understand why the LACIS modeled results by the SMB reached a plateau but not the PIMCA-PINC? Is it the model instrument specific?*

The SBM model is not observed to be instrument dependent as has been previously shown by Peckhaus et al. (2016) where microcline data from a cold stage has successfully been modelled based on Niedermeier et al. (2015) data. Also, already in Niedermeier et al. (2015), SBM calculations based on LACIS measurements were found to agree well with data from Atkinson et al. (2013), another cold stage method. The increase in FF of the PIMCA-PINC data has been reproduced well by the SBM, however, SBM produces a plateau (constant AF as a function of decreasing T) whereas PIMCA-PINC observations show a levelling-off in the slope of the FF (small increase in AF as a function of decreasing T). The reason for this discrepancy is currently unclear and a plateau as modeled by the SBM has also not been observed with similar instrumentation from other previous studies (see p. 9, line 16-17).

The following statement has been added to the manuscript: *"In a recent study by Peckhaus et al. (2017) the SBM was successfully used to reproduce other ice nucleation data from a cold stage experiment. In principle, a contact angle distribution describes the ice nucleation ability of a material, and is then combined with classical nucleation theory in the SBM. Thus the SBM is not believed to be instrument specific."* (p. 9, line 22 ff).

*Specific comments:*
*Page 2, line 3: Do the authors mean: "its importance"?*
The sentence has been changed the following: *"The importance of ice nucleation mechanisms and the properties of aerosol particles acting as so-called ice nucleating particles (INPs),…"* (p. 2, line 4-5).

*Page 2, line 3: I don't think this is the only paper showing this. Please add more references here.*
We agree with the reviewer and have added two more references here (Boucher et al., 2013, Mülmenstädt et al., 2015 (p. 2, line 3).

*Page 2, line 4: Spell-out INP.*
Done.

*Page 2, line 6: Add references after "climate".*
The following references have been added in the suggested location: DeMott et al. (2010) and Phillips et al. (2013). (p. 2, line 7)

*Page 2, line 19: Replace "which" with "what".*
The sentence has been changed as follows: "Which ice nucleation pathways exist and under what conditions they are relevant in the atmosphere is not fully understood, but has been speculated and discussed (e.g., Kanji et al., 2017). (p. 2, line 20-22)

*Page 2, line 20: Add references after "understood".*
We have added a reference as suggested by the reviewer (Vali et al., 2015, Kanji et al., 2017, p. 2, line 5-6).

*Page 2, lines 20-22: I don't understand what the authors want to communicate here. Please clarify it.*
To avoid confusion the following sentence has been deleted from the manuscript*: "However, there is no reason to believe that in the atmosphere, and particularly in mixed-phase clouds, a difference in the freezing mechanism might be relevant, as ice formation in these clouds generally proceeds via the liquid phase."*

*Page 2, line 25: hourly? daily?*
The sentence has been rephrased using *"hour-to-day timescales"* when referring to online ice nucleation measurements (p. 2, line 28).

*Page 2, line 33: DeMott et al., 2016 could be add it to the list.*
The suggested reference has been added (p. 3, line 2).

*Page 3, line 10: Please state what type of CFDC the authors refer to. CSU?*
We have added "CSU-CFDC" to be clear which CFDC we refer to in all relevant locations (p. 3, line 4, 5 13 and 15; p. 19, line 33. The acronym "CFDC" without further distinction is used throughout the manuscript for all continuous flow diffusion chambers and for statements concerning typical CFDCs (including e.g., PINC and/or SPIN).

*Page 3, line 12: Please state what type of mineral dust was used.*
We have now specified in the revised manuscript the sources of the dust used (see p. 3, line 15-16).

*Page 3, line 33: Add references after "activity".*
We have slightly modified the preceding sentence and added a reference (see p. 3, line 35 ff) to read as follows: "*During long-range transport of aerosol particles in the atmosphere, internal mixing with organics and inorganic constituents can cause a temporary or permanent change in the physicochemical properties of the particles and can decrease their ice nucleation activity as discussed in Kanji et al., 2017.*"

*Page 4, line 23: I found the pore size quite big. Is there any chance that small fragments of the pollen grains could go through the filter pores?*
The pore size of the filters is indeed comparably high. However, we like to refer the reader to Pummer et al. (2012) who have investigated the same pollen type as used in this study. They examined the pictures of an electron microscope (dried pollen extracts) to make sure that no submicron particle fragments are left in their solution and the ice active components were much smaller than the filter size. In addition, experiments in this study were performed for size-selected particles in the submicron range, thus any larger particles would not have had any importance on the results.

*Page 4, line 31 and Figure 1: I think it should be "Nuclei" instead of "Nucleus".*
We agree and have made the corresponding changes in the revised manuscript (p. 5, line 8 and Fig. 1 caption).

*Page 5, line 12: "parallel" is out of place.*
The sentence has been rephrased by replacing "*parallel*" by "*simultaneously*". It reads as follows in the revised manuscript: *"Typical particle concentrations during the ice nucleation experiments measured simultaneously with a CPC were 240±70 cm$^{-3}$ for the presented measurements and were diluted to 25-40 cm$^{-3}$ for PIMCA-PINC measurements."* (p. 6, line 8)

*Page 6, line 12: Please state what diameter the authors are referring to.*
We refer to the particle size threshold to distinguish unactivated aerosol particles from ice crystals. The sentence has been changed and reads as follows: *"For data analysis in this study an ice crystal size threshold of 2 µm (diameter) is used."* (p. 7, line 7 ff)

*Page 7, line 29-30: Based on Garimella et al. (2016), the SPIN has a depolarization optical detector. I am wondering why the authors did not use this to discriminate between ice particles and droplets, instead to focus on their size only?*
The reviewer is correct in pointing out that SPIN has a depolarization detector to distinguish between ice particles and droplets. The depolarization signal was not used in the current study due to two reasons:

    a)      To make the comparison of SPIN and PINC data more direct (PINC uses an OPC only measuring particle size).

b) Monodisperse particles were used for the experiments allowing reliable distinguishing of ice from dry particles. The limiting factor for the discrimination from a depolarization detector is the growth of ice crystals that are large enough (which is the case for LACIS and PIMCA-PINC). The depolarization signal of very small ice is not distinct from the dry particle signal and therefore could be prone to errors for smaller ice crystals as expected to be formed in PINC and SPIN in this study.

*Page 8, line 6: Delete Hartmann et al., 2011.*
Change made.

*Page 10, line 25: I am wondering why the authors did not perform an experiment with 500 nm microcline particles as done for kaolinite and Birch pollen.*
The particle types and sizes were chosen based on the following criteria and considering available experiment time of the study:
a) The general selection of particle types and sizes has been made to allow instrument inter-comparison for the maximum range of operational conditions for the instruments to evaluate the instruments performance in comparison with each other (see p. 4, line 4-5).
b) Technical limitations to produce a high enough particle concentration for the measurements with all participating instruments. For example for microcline particles a sufficient particle concentration for the measurement was not achieved for a particle size of 500 nm size-selected particles.

We have added this explanation in methods section and reads as follows: "*The specific particle sizes were chosen to allow measurements in the whole range of detectable frozen/activated fractions for all instruments and for comparison with literature data. A limiting factor for larger particle sizes was the particle generation system, which did not produce a sufficiently high particle concentration for simultaneous measurements with all instruments, thus for example microcline was not tested for sizes larger than 300 nm.*" (p. 6, line 4 ff).

*Page 11, line 1: Add references after "INPs".*
Two references (Archuleta et al., 2005, Welti et al., 2009) have been added to the manuscript (p. 11, line 19).

*Page 11, line 16: "note" is out of place.*
The sentence has been rephrased in the revised manuscript as follows: "*Note, that measurements with PIMCA-PINC using the same particles show a significant reduction in the ice activity with a $T_{50}$ of ~2 K lower for wet generated kaolinite (Fluka) particles for measurements conducted at ETH Zurich in succession to LINC (see Appendix C for more details).*" (p. 11, line 34 ff)

*Page 12, line 18-19: This is a bit unclear. Please clarify it.*
The sentence has been rephrased and now reads as follows: "*This is observed in the PINC data, which indicates that the most active particles are found at 233 K with initial onset of ice formation at $RH_w$ of 82-86% corresponding to $RH_i$ of 121-127%.*" (p. 13, line 14-15)

*Page 14, line 28-32: Given that these experiments differ from the others (i.e., they are not directly comparable), I am wondering if this should be removed for consistency.*
The data on nitric acid treated microcline for the PINC and SPIN instruments are shown here for completeness. All aerosol particles were prepared and produced in the same manner with the same aerosol generation instruments and both data sets were taken individually by each instrument. There are no limiting factors to exclude the data from this study except that measurements were not performed in parallel, from which a difference was not to be expected. The explanation has been revised as follows: "*Note that measurements on nitric acid treated microcline were performed on two different batches of nitric acid treated samples, i.e. PINC and SPIN did not measure in parallel for this*

*aerosol type for which a discrepancy was not expected. It is possible that the observed difference between SPIN and PINC data is based on the ice active material in the PINC batch which may not have been as thoroughly deactivated during acid treatment compared to the batch measured with SPIN."* (p. 15, line 12 ff).

*Page 17, line 11-12 and Figure 7: Add "W" to RH.*
The authors would like to keep "*RH*" without the index "w" in this respect. The reason is that the RH here refers to both, $RH_w$ and/or $RH_i$ (p. 18, line 10-11).

*Page 18, line 3: I don't think this is the only paper showing this. Please add more references here.*
The sentence: *"Recently, CFDCs have often been used for field measurements of INP concentration at water supersaturated conditions to represent immersion freezing (e.g., Kanji et al., 2017 and references therein"* has been deleted from the manuscript.
We rephrased the sentence, which reads as follows in the revised version: *"In many field measurements CFDCs have been used for measurements of INP concentration at water supersaturated conditions (e.g., DeMott et al., 2010, 2016, Tobo et al., 2013, Boose et al., 2016, Lacher et al., 2017) and sometimes are used to represent immersion freezing (e.g., DeMott et al., 2017)."* (p. 19, line 6 ff)

*Table 1. Why is microcline twice here?*
The two lines for microcline have been used to distinguish between 200 nm and 300 nm size-selected aerosol particles. This distinction has been made more clear by adding *"(200 nm)"* and *"(300 nm)"* to the current manuscript (Table 1, first column, rows 1-2).

[revised manuscript text omitted]